# Towards Robust and Scalable Density-based Clustering via Graph Propagation

## Abstract

We present *CluProp*, a novel framework that reimagines varied-density clustering in high-dimensional spaces as a label propagation process over neighborhood graphs. Our approach formally bridges the gap between density-based clustering and graph connectivity, leveraging efficient propagation mechanisms from network science to mitigate the parameter sensitivity inherent in traditional density-based methods. Specifically, we introduce a deterministic density-based propagation strategy to ensure scalable neighborhood identification. The framework is agnostic to the choice of distance metric and exhibits superior performance on large-scale data, processing millions of points in minutes while consistently outperforming existing baselines in accuracy.

## 1. Introduction

Density-based clustering (Campello et al., 2020) identifies clusters as dense regions separated by sparse areas, allowing it to detect arbitrarily shaped clusters and handle noise effectively. Unlike methods like $k$-means (Lloyd, 1982), it is *agnostic* to the choice of distance metric, does not assume spherical clusters or require the number of clusters in advance, making it ideal for complex, real-world data.

Representative density-based clustering methods, such as DBSCAN (Ester et al., 1996) and Density Peak Clustering (DPC) (Rodriguez & Laio, 2014), can be interpreted as *label propagation* mechanisms over a graph defined by local density relationships. In this view, clusters are seeded at high-density points, called core points in DBSCAN or density peaks in DPC, which act as the initial label sources. The labels are then propagated from high-to-low density regions, following the structure of a neighborhood graph, which typically assigns remaining points to the label of its near neighbor with similar densities. This formulation natu-

rally respects the underlying density landscape: labels do not cross low-density regions, allowing the algorithm to discover non-convex clusters and separate noisy or sparse areas. Thus, clustering emerges from how labels flow along high-density paths in the neighborhood graph.

Considering each data point as a node in a graph, DBSCAN or DPC has two primary steps, including (1) constructing an $\varepsilon$-neighborhood or $k$-nearest neighbor ($k$NN) graph to discover density and neighborhood of each node, and (2) propagating cluster labels from dense nodes to sparse nodes.

The first step is the main computational bottleneck as forming these neighborhood graphs requires a worst-case $O(n^2)$ time for a dataset of $n$ points in high dimensions on popular metric distances (Matousek, 1994; Weber et al., 1998). Prior work has tackled this issue by leveraging approximate nearest neighbor search (ANNS) techniques, such as random projections (Schneider & Vlachos, 2017; Xu & Pham, 2024), hashing (Esfandiari et al., 2021; Okkels et al., 2025) or sampling (Viswanath & Babu, 2009; Jang & Jiang, 2019; Jiang et al., 2020), to approximate the graph construction. Other work (de Berg et al., 2019; Gan & Tao, 2017; Mai et al., 2016; Amagata & Hara, 2021; Huang & Ma, 2023) follow the prune-and-bound strategies based on geometric properties on metric spaces to reduce the number of $\varepsilon$-neighborhood queries while forming clusters.

While density-based clustering and its approximate variants are effective for discovering arbitrarily shaped clusters, they struggle on datasets with highly varied densities due to their reliance on global parameters. DBSCAN requires a fixed neighborhood radius ($\varepsilon$) and minimum points ($minPts$), which makes it difficult to capture clusters of different densities simultaneously: a setting that works for dense regions may cause sparse clusters to be missed or merged with noise. Similarly, DPC relies on a cutoff distance ($\varepsilon$) to compute local density and to derive a decision graph to identify cluster centers, but distinguishing true peaks becomes ambiguous when density differences across clusters are extreme. As a result, both algorithms often fail to capture meaningful structures in heterogeneous data, due to their inability to adapt clustering behavior to various local density scales.

Several methods address this issue in datasets with varied densities by incorporating local density estimates derived from $k$NN distances (Du et al., 2019; Yaohui et al., 2017;

[1]Anonymous Institution, Anonymous City, Anonymous Region, Anonymous Country. Correspondence to: Anonymous Author <anon.email@domain.com>.

Preliminary work. Under review by the International Conference on Machine Learning (ICML). Do not distribute.

Durani et al., 2022) or dynamically adjusting the parameter $\varepsilon$ based on the local neighborhood structure (Ankerst et al., 1999; Campello et al., 2015) to overcome the limitation of the global parameter $\varepsilon$ in DBSCAN and DPC. While effective in capturing heterogeneous clusters, these methods often lack scalability, as they rely on the *exact $\varepsilon$*-neighborhood or $k$NN graphs. These operations become computationally intensive on large, high-dimensional datasets.

**Contribution.** We introduce *CluProp*, a scalable and metric-agnostic framework that unifies classical density-based clustering with graph-theoretic paradigms. Instead of relying on rigid connectivity thresholds $\varepsilon$, CluProp utilizes geometric-distance-weighted propagation over neighborhood graphs. This transition enhances clustering accuracy by leveraging efficient modularity-based propagation, such as Louvain (Blondel et al., 2008) or Leiden (Traag et al., 2019), to identify clusters in *approximate $k$NN* graphs across large-scale, high-dimensional datasets with heterogeneous densities.

Since Leiden and Louvain become the primary hurdle of CluProp on large graphs, we propose *DANE* (Density-Aware Neighborhood Expansion), a deterministic algorithm that propagates labels from local density peaks in descending order of density through the neighborhood graph. DANE can be interpreted as a single-linkage-style Potts surrogate without a null baseline, explicitly encoding density-based expansion, and hence linking classical density-based clustering with graph-based propagation. Its simplicity and deterministic nature make it particularly well-suited for high-degree neighborhood graphs where computational efficiency is paramount.

**Scalability.** By leveraging highly optimized implementations for approximate $k$NN graphs (McInnes, 2021) and label propagation (Csardi & Nepusz, 2006), CluProp achieves superior clustering accuracy and runtime performance compared to existing baselines across various distance metrics. On MNIST ($n = 70,000$), CluProp reaches a **90% AMI** score in 20 seconds, significantly outperforming DCN (Yang et al., 2017), a deep learning-based clustering approach that requires over 30 minutes to reach 75% AMI.

The scalability of our framework is further demonstrated on MNIST8M ($n = 8,100,000$). Using DANE on approximate $k$NN graphs, CluProp completes clustering in under 15 minutes on a single workstation, achieving an **80% NMI**. In contrast, kernel $k$-means (Wang et al., 2019) achieves only **41% NMI** despite being executed on a Spark-based supercomputing cluster.

## 2. Preliminaries

**Density-based Clustering (DBSCAN)** (Ester et al., 1996) identifies clusters as connected regions of high densities. Given a distance measure $d(\cdot, \cdot)$, DBSCAN is parameter-

ized by a radius $\varepsilon$ and a density threshold $minPts$. For each point $\mathbf{x}_i \in \mathbf{X}$, DBSCAN performs an $\varepsilon$-*range query* $B_\varepsilon(\mathbf{x}_i) = \{\mathbf{x}_j \in \mathbf{X} \,|\, d(\mathbf{x}_i, \mathbf{x}_j) \leq \varepsilon\}$, and classifies $\mathbf{x}_i$ as a *core* point if $|B_\varepsilon(\mathbf{x}_i)| \geq minPts$; otherwise, it is considered a *non-core* point.

Clusters are formed by recursively connecting core points within mutual $\varepsilon$-neighborhoods. This process can be interpreted as a graph label propagation where edges link core points within $\varepsilon$-distance. While effective for uniform-density clusters, the reliance on global parameter $\varepsilon$ and $minPts$ limits its performance on varied-density datasets.

**Density Peak Clustering (DPC)** (Rodriguez & Laio, 2014) addresses DBSCAN's sensitivity to global parameters by incorporating a more adaptive density estimation. Given a cutoff distance $\varepsilon$, DPC computes a local density score $\rho_i = |B_\varepsilon(\mathbf{x}_i)|$. DPC identifies cluster centers as points with high local density $\rho_i$ and large distance $\delta_i$ to any point of higher density, where $\delta_i = \min_{j:\rho_j > \rho_i} d(\mathbf{x}_i, \mathbf{x}_j)$. Once cluster centers are selected, labels are propagated by assigning each remaining point to the same cluster as its nearest neighbor of higher density, effectively following the ascending density gradient. Similar to DBSCAN, DPC's performance still hinges on the choice of $\varepsilon$, and a poorly chosen cutoff leads to suboptimal clustering results.

**Propagation in Graph.** LPA (Raghavan et al., 2007) is among the first graph clustering and community detection, valued for its simplicity, efficiency, and ability to uncover densely connected structures without requiring the number of clusters in advance. Louvain (Blondel et al., 2008) and Leiden (Traag et al., 2019) are widely used label propagation algorithms that identify *modular* structures in large-scale graphs. Both approaches greedily optimize modularity (Newman, 2006) by moving each node to the neighboring community that yields the highest modularity gain, followed by aggregation of communities into super-nodes and repeating the process hierarchically.

Modular-based propagation algorithms naturally support weighted graphs by incorporating edge weights into the modularity computation. This allows them to capture fine-grained community structures where edge weights encode pairwise similarities of points, particularly beneficial in $k$NN graphs derived from high-dimensional datasets.

## 3. A Baseline via Asymptotic Equivalence of DBSCAN and Graph Connectivity

This section establishes theoretical conditions under which DBSCAN cluster structures, defined via $\varepsilon$-neighborhood graphs with varying $\varepsilon$ values, can be recovered through the connectivity of *mutual $k$NN* graphs. These conditions enable the use of $k$NN graph to approximate the behavior of DBSCAN under non-uniform densities by adapting neigh-

borhood size implicitly through local point distributions. This connection lays the groundwork for recent approaches that recover density-based clustering using $k$NN graph variants (Yaohui et al., 2017; Du et al., 2019; Li et al., 2024; Sun et al., 2025), providing both practical scalability and theoretical justification.

Given a dataset $\mathbf{X}$ of $n$ points $\mathbf{x}_i \in \mathbb{R}^d$, the *mutual $k$NN graph* constructed on $\mathbf{X}$ is symmetric and has an edge $(\mathbf{x}_i, \mathbf{x}_j)$ if $\mathbf{x}_i \in k\text{NN}(\mathbf{x}_j) \wedge \mathbf{x}_j \in k\text{NN}(\mathbf{x}_i)$. We will use the following result to prove our main result regarding the equivalence of DBSCAN and $k$NN graph connectivity.

**Theorem 3.1** (Connectivity of Mutual $k$NN Graph (Brito et al., 1997)). *Let $\mathbf{X} = \{\mathbf{x}_1, \ldots, \mathbf{x}_n\} \subset \mathbb{R}^d$ be i.i.d. samples drawn from a probability density function $f : \mathbb{R}^d \to \mathbb{R}_+$ with compact support. Let $\mathcal{L}_{f_0} = \{\mathbf{x} \in \mathbf{X} : f(\mathbf{x}) \geq f_0\}$ be a connected superlevel set of the density. Construct the mutual $k$-nearest neighbor graph $G_k$ on $\mathbf{X}$, and let $G_k[\mathcal{L}_{f_0}]$ be the subgraph induced by points in $\mathcal{L}_{f_0}$.*

*Then, if*

$$k \geq c \log n$$

*for some constant $c > 0$, it holds that with high probability as $n \to \infty$, the graph $G_k[\mathcal{L}_{f_0}]$ is connected.*

We assume that the following hold:

1. (Smoothness) $f$ is Lipschitz continuous with constant $\alpha > 0$.

2. (Cluster Structure) Clusters are defined as connected components of the superlevel set $\{\mathbf{x} : f(\mathbf{x}) \geq f_0\}$ for some threshold $f_0 > 0$.

3. (Cluster Separation) Any path from one cluster to another must pass through a region where $f(\mathbf{x}) < f_0$.

4. (Sampling) Let $d_k(\mathbf{x})$ be the $k$NN distance of $\mathbf{x}$ given $k = o(n)$ and $k \to \infty$ when $n \to \infty$.

The smoothness assumption ensures that nearby points in the same cluster will have similar density. The second and third ones ensure that the true clusters are located in high-density areas, and are separated away from noise regions with sufficiently low density. The last assumption ensures that the $k$NN-based density estimator $\widehat{f}(\mathbf{x})$ converges to the true density $f(\mathbf{x})$ almost surely. These assumptions are widely used to analyze density-based clustering algorithms (Chaudhuri et al., 2014; Jiang, 2017; Jiang et al., 2020; Xu & Pham, 2024).

We define a *varied-density DBSCAN*, called $DBSCAN_k^*$, where we set $minPts = k$ and *sequentially* run DBSCAN using a series of increasing density thresholds $\varepsilon_\ell < \varepsilon_{\ell-1} < \ldots < \varepsilon_1$, recovering density-based clusters $L_\ell, \ldots, L_1$ in order from the densest to the sparsest. At each stage, the algorithm operates on the set of non-core points from the previous run, allowing cluster discovery at progressively

lower density levels. The following result establishes that the cluster structure induced by $DBSCAN_k^*$ can be recovered via connectivity in a mutual $k$NN graph. The proof using four assumptions above is left in the appendix.

**Theorem 3.2** (DBSCAN Cluster Recovery via Mutual $k$NN Connectivity). *Let $\mathbf{X} = \{\mathbf{x}_1, \ldots, \mathbf{x}_n\} \subset \mathbb{R}^d$ be i.i.d. samples drawn from a Lipschitz continuous density function $f$, and let $\mathcal{L}_{f_0} = \{\mathbf{x} \in \mathbb{R}^d : f(\mathbf{x}) \geq f_0\}$ be a union of $L$ disjoint, compact, connected components $\mathcal{L}_{f_i} = \{\mathbf{x} \in \mathbb{R}^d : f(\mathbf{x}) \geq f_i\}$, where $f_0 \leq f_1 < \ldots < f_\ell$, separated by low-density regions where $f(\mathbf{x}) < f_0$.*

*Construct the mutual $k$-nearest neighbor graph $G_k$ on $\mathbf{X}$, where $k \geq c \log n$ for a sufficiently large constant $c > 0$. Run $DBSCAN_k^*$ with $\ell$ density thresholds $\varepsilon_i = (k/nV_d f_i)^{1/d}$ where $V_d$ denotes the volume of the unit ball in $\mathbb{R}^d$.*

*Then, with high probability as $n \to \infty$:*

- *The mutual $k$NN graph $G_k$ is connected within each high-density component $\mathcal{L}_{f_i}$ corresponding to the cluster $L_i$.*

- *The clusters of $DBSCAN_k^*$ over various values of $\varepsilon$ can be recovered asymptotically by identifying connected components of core points in $G_k$.*

Theoretically, Theorem 3.2 implies that a simple label propagation procedure, starting from a randomly selected high-density point and propagating its label to all reachable points in the mutual $k$NN graph. This process recovers the varied-density clustering structure produced by $DBSCAN_k^*$ with high probability. This theoretical correspondence facilitates a parsimonious clustering baseline that relies exclusively on $k$NN graph connectivity, bypassing the need for $\varepsilon_i$-neighborhood graphs where the values of $\varepsilon_i$ are often difficult to tune.

# 4. Clustering via Graph Propagation

We first introduce CluProp, a density-based clustering framework that leverages *approximate $k$NN graphs* and modularity-driven propagation to identify latent clusters. As modularity-based propagation becomes the primary hurdle on large-scale datasets, we propose DANE (*Density-Aware Neighborhood Expansion*), an efficient deterministic propagation alternative that simulates density-based expansion while significantly reducing computational overhead.

## 4.1. CluProp: A Density-based Clustering Framework

While the simple label propagation algorithm could be applied to a mutual $k$NN graph to form clusters, this approach faces two significant challenges in practice. First, capturing heterogeneous densities often requires a prohibitively large $k$, making the construction of an exact mutual $k$NN graph

**Algorithm 1** CluProp: Clustering via graph propagation

**Require:** Dataset $\mathbf{X}$, $k$, distance $d(\cdot, \cdot)$
1: For each point, find approximate $k$NN by any ANNS library
2: Build a weighted symmetric $k$NN graph $G_k$
3: **Return** Communities of $G_k$ by a graph propagation (e.g., Leiden or Louvain)

---

computationally infeasible. Second, propagation on a mutual $k$NN graph typically fails to utilize the pairwise distance information already computed during graph construction, discarding valuable local geometric data.

To address these limitations, we utilize a *weighted symmetric* $k$NN graph, where an edge $(\mathbf{x}_i, \mathbf{x}_j)$ exists if $\mathbf{x}_i \in k\text{NN}(\mathbf{x}_j) \lor \mathbf{x}_j \in k\text{NN}(\mathbf{x}_i)$. Each edge is assigned a weight proportional to $d(\mathbf{x}_i, \mathbf{x}_j)$. The symmetric version not only preserves the essential graph connectivity but also enhances robustness against fragmentation in regions of varied density, a common failure mode for sparser mutual $k$NN graphs.

Algorithm 1 shows the CluProp framework, which utilizes *weight-aware modularity-based* propagation algorithms, including Louvain (Blondel et al., 2008) and Leiden (Traag et al., 2019), applied to an *approximate* weighted symmetric $k$NN graph. Modularity quantifies the quality of a cluster by comparing the observed intra-cluster edge weights to the expected connectivity under a random null model. In particular, for a weighted graph with edge weights $w_{ij}$, modularity is defined as:

$$M = \frac{1}{2m} \sum_{i,j} \left( w_{ij} - \frac{s_i s_j}{2m} \right) \mathbf{1}[L_i = L_j],$$

where $s_i = \sum_j w_{ij}$ denotes the strength (weighted degree) of node $i$, $\sum_{i,j} w_{ij} = 2m$ represents the total edge weight.

In high dimensional spaces, the incorporation of edge weights is vital for capturing local density variations. By weighting edges according to pairwise similarities, the modularity objective becomes sensitive to the underlying data geometry. High-weight edges between mutual neighbors increase $M$ when assigned to the same cluster, whereas strong edges between clusters are heavily penalized. This weighting scheme effectively mitigates the "chaining effect" prevalent in *approximate* unweighted graphs, where sparse, low-similarity connections (noise) might incorrectly bridge two distinct, high-density regions.

By leveraging efficient approximate nearest neighbor search (ANNS) libraries, CluProp achieves a high-performance profile that balances computational scalability with structural fidelity. Although the framework remains dependent on the neighborhood size $k$ and the precision of the $k$NN graph construction, it significantly enhances robustness by elimi-

**Algorithm 2** Density-aware Neighborhood Expansion

**Require:** $k$, weighted graph $G$, density estimates $1/d_k(\cdot)$, $N(\mathbf{x}) = \{\mathbf{x}' \in \mathbf{X} \mid \exists\ (\mathbf{x}, \mathbf{x}') \in G\}$
1: **Initialization**: Sort all points by descending density; initialize labels $L \leftarrow -1$
2: **for** each unlabeled point $\mathbf{x}$ in sorted density order **do**
3:     **Seed Cluster**: Increment cluster count and assign $L[\mathbf{x}]$; initialize priority queue $Q$
4:     **Initialize Frontier**: For all unlabeled neighbors $\mathbf{x}' \in N(\mathbf{x})$, push to $Q$ with priority $d(\mathbf{x}, \mathbf{x}') + d_k(\mathbf{x}')$ and predecessor $\mathbf{x}$
5:     **while** $Q$ is not empty **do**
6:         **Extract Candidate**: Pop $(\mathbf{y}, \mathbf{p})$ with minimum priority (where $\mathbf{p}$ is the predecessor of $\mathbf{y}$)
7:         **if** $\mathbf{y}$ is unlabeled **then**
8:             **Connectivity Check**: If a point $\mathbf{y}' \in k\text{NN}(\mathbf{y})$ is already in $\mathbf{p}$'s cluster, set $L[\mathbf{y}] \leftarrow L[\mathbf{p}]$.
9:             **Boundary Handling**: Otherwise, $\mathbf{y}$ represents a density shift; initiate a new cluster for $\mathbf{y}$.
10:            **Expand Frontier**: For all unlabeled neighbors $\mathbf{y}' \in N(\mathbf{y})$, push to $Q$ with priority $d(\mathbf{y}, \mathbf{y}') + d_k(\mathbf{y}')$ and predecessor $\mathbf{y}$.
11:         **end if**
12:     **end while**
13: **end for**
14: **Return** Cluster labels $L$

---

nating the reliance on a globally fixed $\varepsilon$ threshold. This shift from rigid distance-based constraints to adaptive graph propagation enables CluProp to navigate heterogeneous densities more effectively, delivering substantial gains in clustering quality across several distance metrics without sacrificing the efficiency required for massive datasets.

### 4.2. DANE: Density-aware Neighborhood Expansion

While CluProp demonstrates high efficiency and accuracy, modularity-based propagation tends to converge slowly on datasets with millions of points, particularly when larger values of $k$ are required to maintain the graph connectivity. To address this, we propose replacing the computationally intensive modularity optimization of the Leiden and Louvain algorithms with DANE: a high-speed, greedy heuristic tailored specifically for density-based clustering.

DANE initiates clustering from the highest-density peaks and propagates labels to proximate, lower-density neighbors along descending density gradients. DANE can be interpreted as a principled method for extracting clusters from the OPTICS-style dendrogram, where cluster formation begins from the deepest valleys and expands through density-reachable paths. Empirically, DANE bypasses the iterative overhead of modularity optimization, running orders of magnitude faster than Leiden or Louvain while maintain-

ing competitive clustering accuracy.

Algorithm 2 outlines the procedure of DANE with two inputs: a parameter $k$ and a weighted graph $G$. It first initiates clustering from the highest-density point $\mathbf{x}$, where density is estimated using the inverse $k$NN distance, $1/d_k(\mathbf{x})$. It then propagates the label $L[\mathbf{x}]$ to unlabeled neighbors $\mathbf{x}'$, prioritizing candidates based on a *composite* score: $d(\mathbf{x}, \mathbf{x}') + d_k(\mathbf{x}')$. This priority function favors nearby points (small $d(\mathbf{x}, \mathbf{x}')$) with high local density (small $d_k(\mathbf{x}')$), ensuring that label propagation respects both spatial proximity and density gradients.

A point $\mathbf{x}'$ is eligible to receive a label of its predecessor $\mathbf{x}$ if one of $k$NN($\mathbf{x}'$) has already been assigned the label $L[\mathbf{x}]$. As we process $\mathbf{x}'$ using the priority score $d(\mathbf{x}, \mathbf{x}') + d_k(\mathbf{x}')$ where $\mathbf{x}$ is the labeled predecessor, $\mathbf{x}'$ is only labeled by its closest labeled predecessor of higher density.

This two-fold prioritization, favoring both descending density and spatial proximity, ensures that label propagation follows the intrinsic structure of the data: *from high-to-low density regions* and *from near-to-far neighbors*. This disciplined propagation strategy mitigates the risk of label diffusion across low-density gaps, enabling clusters to grow coherently from dense cores. A detailed pseudocode and illustration of DANE are provided in the appendix.

**Time complexity.** Given the weighted symmetric $k$NN graph input, DANE is deterministic and runs in $O(nk \cdot \log(nk))$ time as the size of priority queue $Q$ is bounded by $O(nk)$. The superior computational efficiency of DANE allows for the use of denser neighborhood graphs (larger $k$) on massive datasets, facilitating higher structural resolution without the prohibitive runtime associated with modularity-based propagation.

### 4.3. Potts-model view of DANE

For a weighted graph with edge weights $w_{ij}$ and the label of node $i$, $L_i \in \{1, \dots, \ell\}$, a broad class of community detection methods can be expressed through a Potts energy of the form (Traag et al., 2011):

$$\mathcal{E}(L) = -\sum_{i,j} A_{ij} \mathbf{1}[L_i = L_j] + \sum_{i,j} B_{ij} \mathbf{1}[L_i = L_j] \,, \quad (1)$$

where $A_{ij}$ encodes attractive interactions and $B_{ij}$ specifies a null or baseline model.

**Modularity.** In modularity maximization, $A_{ij} = w_{ij}$ and $B_{ij} = \frac{s_i s_j}{2m}$, where $s_i = \sum_j w_{ij}$ and $m = \frac{1}{2} \sum_{i,j} w_{ij}$. The modularity gain of moving node $i$ to cluster $C$ is

$$\Delta Q(i \to C) = \sum_{j \in C} w_{ij} - \frac{s_i \operatorname{vol}(C)}{2m} \,, \quad (2)$$

corresponding to greedy coordinate descent on (1).

**Label propagation (LPA).** LPA can be viewed as a constant Potts model with $B_{ij} = \gamma$. Setting $A_{ij} = w_{ij}$ yields the local gain:

$$\Delta_{\text{LPA}}(i \to C) = \sum_{j \in C} w_{ij} \,. \quad (3)$$

**DANE.** DANE further restricts the Potts interaction to nearest-neighbor attachment. For node $i$, the gain of assigning $i$ to cluster $C$ is defined as

$$\Delta_{\text{DANE}}(i \to C) = \max_{j \in C \cap \text{kNN}(i)} w_{ij} \,. \quad (4)$$

To realize (4), we can see that DANE only assigns the predecessor label $L[\mathbf{x}_j]$ to $\mathbf{x}_i$ if one of $k$NN($\mathbf{x}_i$) had label $L[\mathbf{x}_j]$, and greedily processes $\mathbf{x}_i$ with a priority score $d(\mathbf{x}_i, \mathbf{x}_j) + d_k(\mathbf{x}_i)$, equivalent to $\max_{j \in C} w_{ij}$ in (4).

DANE replaces the sum-based Potts interaction with a max-based surrogate, corresponding to a single-linkage-style objective that is invariant to cluster size. Combined with density-ordered processing via the $k$NN radius, DANE induces a constrained Potts optimization that aligns with density-based connectivity rather than global modularity.

**Algorithmic distinction.** DANE inherits its update order from density-based clustering: points are processed in a density-respecting order induced by the $k$NN radius, enforcing monotone label propagation from high-density cores to lower-density regions, as in DBSCAN and DPC. This ordering is an essential part of the algorithm and prevents propagation across density valleys. In contrast, modularity-based methods are defined as order-agnostic global optimization problems, where the update sequence does not encode density or expansion direction.

## 5. Experiment

We implement DANE in C++ and compile with `g++ -O3 -std=c++17 -fopenmp -march=native`. We conducted experiments on Ubuntu 20.04.4 with an AMD Ryzen Threadripper 3970X 2.2GHz 32-core processor with 128GB of DRAM. We use highly-optimized industry libraries, including **PyNNDescent** (McInnes, 2021) to construct approximate $k$NN graphs and **igraph** (Csardi & Nepusz, 2006) for modularity-based propagation, in our CluProp framework. We use 32-thread PyNNDescent to construct $k$NN graphs across various distance metrics. The implementation of graph propagation algorithms, including Leiden, Louvain, LPA, and DANE, do not support multi-threading.

Our empirical evaluations on the clustering quality compared to the ground truth (i.e. data labels) will verify:

- CluProp outperforms existing baselines in accuracy and runtime across several distance metrics, despite the rigorous parameter tuning of the comparative methods.

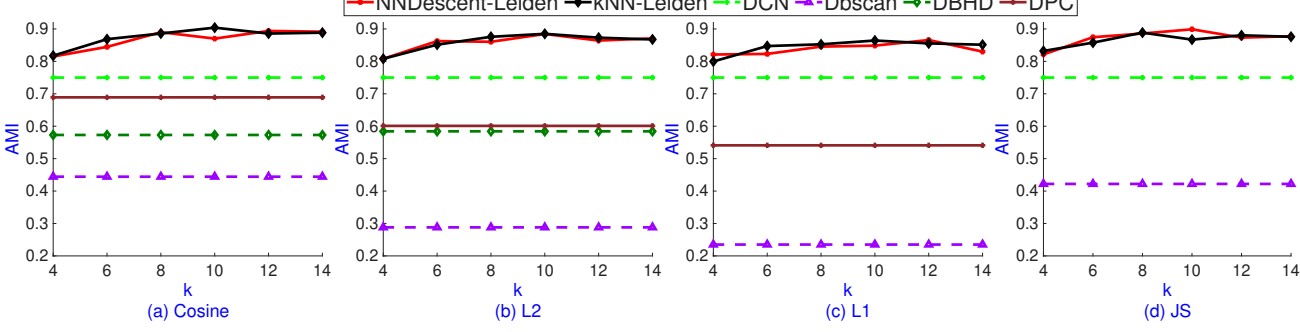

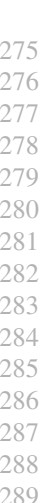

Figure 1. The comparison of AMI provided by CluProp (NNDescent and exact $k$NN for graph constructions with Leiden) and the maximum AMI scores provided by other clustering baselines across various parameter settings on Mnist.

Table 1. The AMI and runtime of CluProp (NNDescent takes 15s and Leiden takes 5s) with $k = 8$ and other clustering baselines with cosine on Mnist. We report the maximum AMI/ARI scores of compared baselines across various parameter settings.

| Alg. | CluProp | sDbscan | Dbscan | Optics | hDbscan | DPC | DBHD | SpectACl | $k$-means | DCN |
|------|---------|---------|--------|--------|---------|-----|------|----------|-----------|-----|
| AMI (%) | **89 ± 1** | 42 ± 4 | 43 | 6 | 31 | 69 | 56 | 80 | 51 | 75 |
| ARI (%) | **87 ± 1** | 9 | 9 | 0 | 5 | 54 | 18 | 70 | 38 | 62 |
| # clu | **12 ± 1** | 29 ± 6 | 7 | 47 | 9 | 12 | 118 | – | – | – |
| Time | 20s | 3s | 34s | 25 min | 1 hour | 5 min | > 2 hour | 15s | 1s | 37 min |

- DANE achieves orders of magnitude higher throughput than the Leiden and Louvain algorithms. CluProp with DANE maintains superior performance in accuracy and runtime on datasets exceeding millions of points compared to existing baselines.

We use popular metrics, including AMI, NMI, ARI, CC (Gösgens et al., 2021) to measure the clustering quality. We mainly report AMI scores in the paper. Results on other metrics are left in the appendix. We conduct experiments on three popular data sets: Mnist ($n = 70,000$, $d = 784$, # clusters = 10), Pamap2 ($n = 1,770,131$, $d = 51$, # clusters = 18), and Mnist8m ($n = 8,100,000$, $d = 784$, # clusters = 10). All results are the average of 5 runs of the algorithms.

Our competitors include (1) representative density-based clustering algorithms that do not need the predefined number of clusters, including DBSCAN (Ester et al., 1996), DPC (Rodriguez & Laio, 2014), HDBSCAN (Campello et al., 2015), OPTICS (Ankerst et al., 1999), DBHD (Durani et al., 2022), sngDBSCAN (Jiang et al., 2020), sDB-SCAN (Xu & Pham, 2024); and (2) representative clustering algorithms that need a predefined number of clusters, including $k$-means, kernel $k$-means (KKM), spectral clustering (SC), SpectACl (Hess et al., 2019), deep learning-based DCN (Yang et al., 2017). Details of tuned parameter values of these algorithms are in the appendix.

PyNNDescent implements NNDescent (Dong et al., 2011), a well-known *iterative* algorithm to construct $k$NN graph

by refining an initial $k$NN graph based on the principle that neighbors of a point are likely to share neighbors. PyNNDescent initializes the graph using an ensemble of Random Projection Trees (RPT) as the convergence of NNDescent highly depends on the quality of the initial graph. Therefore, the two parameters, number of RPTs and number of iterations, govern the running time and final graph quality, and hence the performance of CluProp. Fortunately, both Leiden/Louvain and DANE are not very sensitive to the graph outputs of PyNNDescent with several configurations.

### 5.1. An ablation study on Mnist

This subsection studies the performance of CluProp that uses PyNNDescent to construct $k$NN graph and modularity-based propagation (Leiden and Louvain) to form clusters. While CluProp does not need much parameter tuning given the $k$NN graph, other competitors require a careful selection of parameter values to achieve reasonable accuracy. We detail the process of selecting parameter values of competitors in the appendix.

We report representative results on Leiden. The results of Louvain are very similar and left in the appendix. PyNNDescent uses 8 RPT for graph initialization, and 5 NNDescent iterations. The graph construction runs within 15 seconds.

**AMI scores of CluProp (PyNNDescent and exact $k$NN with Leiden) and other clustering baselines.** Figure 1 shows the AMI scores over a wide range of $k$ on cosine, L2,

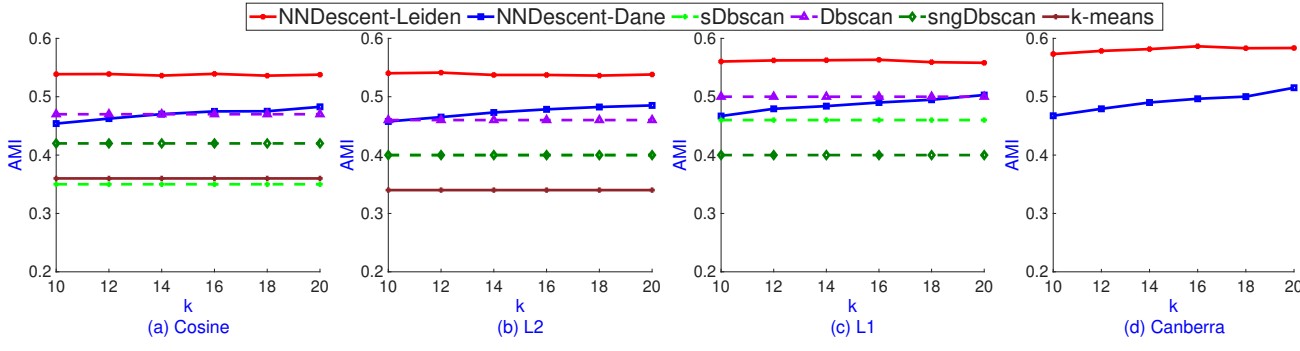

*Figure 2.* The comparison of AMI provided by CluProp (NNDescent for graph constructions with Leiden and DANE) and the maximum AMI scores provided by other clustering baselines across various parameter settings on Pamap2.

*Table 2.* The AMI and runtime of CluProp with $k = \{10, 20\}$ and other clustering baselines with L2 on Pamap2. PyNNDescent takes **{0.6, 0.8} min**, respectively. We report the maximum AMI scores of compared baselines across various parameter settings.

| *Alg.* | $k = 10$ | | | $k = 20$ | Dbscan | sDbscan | sngDbscan | $k$-means |
|---|---|---|---|---|---|---|---|---|
| | Leiden | Louvain | DANE | DANE | | | | |
| AMI | **54** | **54** | 46 | **49** | 47 | $44 \pm 2$ | 38 | 34 |
| Propagation time (min) | 1.8 | 2.5 | **5 sec** | **7 sec** | – | – | – | – |
| Total time (min) | 2.4 | 3.1 | **0.6** | **0.9** | 30 | 1 | 3 | 3 sec |

L1, and Jensen-Shannon (JS) distances. For other selected clustering algorithms, including DBSCAN, DBHD, DPC, DCN, we report the *highest* AMI scores after several runs with different values of parameters. It is clear that our proposed CluProp baselines outperform other clustering competitors on several distance measures. In particular, they achieve at least 10%, 15%, 20%, and 45% higher AMI scores compared to DCN, DPC, DBHD, and DBSCAN, respectively.

While graph construction is nearly six times faster, the use of PyNNDescent yields clustering accuracy comparable to that of an exact $k$NN graph. This highlights the efficacy of our weighted modularity-based propagation, which captures subtle local density variations to enhance structural fidelity and offset the minor approximations in the underlying graph.

**Performance comparison between CluProp with other clustering baselines.** Table 1 provides a systematic evaluation of CluProp (PyNNDescent and Leiden) regarding runtime and accuracy on cosine distance compared to existing baselines, including density-based clustering without predefined number of clusters (sDBSCAN, DBSCAN, OPTICS, HDBSCAN DPC, DBHD) and other clustering with predefined number of clusters (SpecACl, $k$-means, DCN). CluProp offers superior performance in both time and accuracy compared to studied baselines. It runs in 20 seconds, slower than sDbscan and $k$-means but achieve nearly 40% higher AMI. Regarding ARI scores, CluProp gives 17%, 25%, 33%, and more than 50% higher than SpectACl, DCN, DPC, and the others, respectively. CluProp also runs signifi-

cantly faster than the others, $15\times$ and $110\times$, compared to DPC and DCN, respectively.

### 5.2. Experiments on million-point datasets

While increasing the neighborhood size $k$ and enhancing the $k$NN graph quality generally improve clustering accuracy of CluProp, such configurations impose a significant computational burden on both graph construction and modularity-based propagation when scaling to million-point datasets. We demonstrate that even with a standard PyNNDescent configuration, modularity-based algorithms remain the primary hurdle within the CluProp framework; conversely, DANE substantially reduces execution time without compromising clustering accuracy. To show the scalability of CluProp, we use PyNNDescent with 8 RPT, and 1 NNDescent iteration to construct $k$NN graphs within a few minutes.

**Pamap2.** Figure 2 shows that CluProp (Leiden and DANE) offers consistently higher AMI scores than the highest AMI provided by other density-based clustering baselines on cosine, L2, and L1. We also observe that Canberra is a relevant metric for Pamap2 as it reaches nearly 60% AMI score while the other popular metrics, such as L1, L2, cosine, only return up to 55%. It verifies the advantage of being agnostic to distance metrics of CluProp, supporting all distance measures provided by PyNNDescent. We note that both Leiden and DANE give stable accuracy on different $k$NN graphs provided by PyNNDescent with different configurations, as shown in the appendix.

*Table 3.* The AMI and runtime of CluProp with $k = \{50, 80\}$ and other clustering baselines with cosine on Mnist8m. PyNNDescent takes **{8.2, 11} min**, respectively. We report the maximum AMI/NMI scores of compared baselines across various parameter settings.

| *Alg.* | $k = 50$ | | | $k = 80$ | sDbscan-1NN | sDbscan | sngDbscan | Kernel |
|---|---|---|---|---|---|---|---|---|
| | Leiden | Louvain | DANE | DANE | | | | $k$-means |
| AMI (%) | **81** | **81** | $72 \pm 2$ | $\mathbf{80 \pm 2}$ | 38 | 32 | 26 | – |
| NMI (%) | **81** | **81** | $72 \pm 2$ | $\mathbf{80 \pm 2}$ | 38 | 32 | 26 | 41 |
| Prop. time (min) | 57 | 44 | **0.9** | **1.1** | – | – | – | – |
| Total time (min) | 65 | 53 | **9** | **12** | 21 | 16 | 42 | 15 |

*Table 4.* The AMI and runtime in minutes of CluProp with DANE ($k = 80$) across several configurations of PyNNDescent on Mnist8m using cosine.

| *Time* | 8 trees 2 iters | 16 trees 1 iter | 8 trees 1 iter | 4 trees 1 iter |
|---|---|---|---|---|
| NNDescent | 24.1 | 15.4 | 11 | 9.7 |
| DANE | 1.1 | 1.1 | 1.1 | 1.1 |
| Total | 25.2 | 16.5 | 12.1 | 10.8 |
| AMI (%) | $81 \pm 0$ | $81 \pm 2$ | $80 \pm 2$ | $77 \pm 4$ |

Table 2 presents the representative runtime between CluProp with Leiden, Louvain, and DANE and other density-based baselines on L2. Similar results are observed across all studied metrics. Again, CluProp demonstrates the superior performance in terms of accuracy and runtime compared to other density-based competitors. Though DANE gives an 9% AMI less than Leiden on $k = 10$, it indeed runs several orders of magnitude faster. Since the graph construction with $k = 10$ is within 0.6 min, modularity-based propagation becomes the hurdle of CluProp. By applying DANE on a larger graph with $k = 20$, we can approximately reach AMI scores provided by Leiden but with a faster runtime. This advantage of DANE over Leiden and Louvain will be further highlighted on Mnist8m.

**Mnist8m.** We study cosine distance, and report the results of recent scalable density-based clustering, including sDbscan, sDbscan-1NN, sngDbscan, and the kernel $k$-means (Wang et al., 2019) run on a supercomputer with 32 nodes. As modularity-based propagation take significant time to run on larger $k$, we report the accuracy on $k = 50$. Results on smaller values of $k$ are in the appendix. Since DANE runs very fast compared to Leiden/Louvain, we run CluProp with DANE with higher degree graphs, and report its performance on $k = 80$.

Table 3 shows the comparison between Leiden/Louvain and DANE in terms of accuracy and runtime on Mnist8m. As DANE runs several order of magnitude faster than Leiden/Louvain, it achieves similar accuracy with $k = 80$ but offers 5 times speedup. CluProp with DANE outper-

forms non-propagation-based baselines in both accuracy and speed. It achieves NMI of 80%, compared to 41% of kernel $k$-means running on a supercomputer, and 32% of sDbscan running on a similar machine configuration.

Table 4 shows the stability in accuracy of DANE across several configurations of PyNNDescent, each lead to different quality of $G_k$. We can see that improving the quality of graph by increasing number of NNDescent iterations or number of RPT will reduce the variance of the accuracy. DANE demonstrates significant robustness to graph quality; even when using a coarse $k$NN approximation (4 RPTs, 1 NNDescent iteration), it yields 77% AMI and surpasses the accuracy of all established baselines.

## 6. Conclusion and Future Work

Our work bridges the gap between density-based clustering and graph connectivity by presenting a theoretically grounded and computationally efficient framework that scales to modern high-dimensional datasets. By unifying approximate $k$NN graph construction with label propagation, our distance-agnostic approach provides a robust and scalable solution for clustering data with complex, heterogeneous structures. In particular, our novel density-aware neighborhood expansion method combined with advanced ANNS solvers delivers strong empirical performance on million-point datasets.

A promising future work is the adaptation of DANE to general graph clustering. In such settings, the notion of density could transition from geometric proximity to topological metrics, such as core numbers (Batagelj & Zaversnik, 2003) or local conductance (Andersen et al., 2006), allowing the framework to partition non-spatial relational networks.

Another important direction is to develop a rigorous theoretical analysis of DANE within the Potts-model framework by incorporating a null baseline, thereby bridging density-ordered, max-based propagation with sum-based modularity objectives. Such an analysis could yield hybrid propagation schemes with formal guarantees that combine the robustness of density-based expansion with the expressiveness of null-model-based formulations.

## Impact Statement

This paper presents work whose goal is to advance the field of Machine Learning. There are many potential societal consequences of our work, none which we feel must be specifically highlighted here.

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

# A. Detailed Pseudocode of DANE

---

**Algorithm 3** DANE: Density-Aware Neighborhood Expansion

---

**Require:** $\mathbf{X}$, $k$, weighted graph $G$, local density estimates $1/d_k(\mathbf{x})$ (inverse of $k$NN distance), $N(\mathbf{x}) = \{\mathbf{x}' \in \mathbf{X} \mid \exists\,(\mathbf{x}, \mathbf{x}') \in G\}$

**Ensure:** Disjoint cluster assignments $L$

1: **// Phase I: Initialization and Sorting**
2: Initialize all labels $L[\mathbf{x}] \leftarrow$ UNLABELED and reachability distance $R[\mathbf{x}] \leftarrow \infty$ for all $\mathbf{x} \in \mathbf{X}$
3: Sort points in $\mathbf{X}$ by descending density $1/d_k(\mathbf{x})$ to prioritize density peaks
4: Initialize an empty priority queue $Q$ and cluster counter $C \leftarrow 0$
5: **// Phase II: Global Traversal of Density Peaks**
6: **for** each point $\mathbf{x} \in \mathbf{X}$ in sorted density order **do**
7:    **if** $L[\mathbf{x}] \neq$ UNLABELED **then**
8:       **continue** {Skip points already assigned a label}
9:    **end if**
10:   **// Phase III: Cluster Seeding**
11:    $C \leftarrow C + 1$
12:    $L[\mathbf{x}] \leftarrow C$
13:    **for** each neighbor $\mathbf{x}' \in N(\mathbf{x})$ **do**
14:       **if** $L[\mathbf{x}'] =$ UNLABELED $\wedge\, d(\mathbf{x}, \mathbf{x}') < R[\mathbf{x}']$ **then**
15:          $R[\mathbf{x}'] \leftarrow d(\mathbf{x}', \mathbf{x})$ {Update reachability distance from the closer peak $\mathbf{x}$}
16:          $priority \leftarrow R[\mathbf{x}'] + d_k(\mathbf{x}')$
17:          $Q.\text{insert}(\mathbf{x}', \text{source} = \mathbf{x}, priority)$
18:       **end if**
19:    **end for**
20:   **// Phase IV: Deterministic Cluster Expansion**
21:    **while** $Q$ is not empty **do**
22:       $(\mathbf{y}, \mathbf{p}) \leftarrow Q.\text{extract\_min}()$ {$\mathbf{p}$ is the predecessor of $\mathbf{y}$}
23:       **// Verify density-connectivity to predecessor p**
24:       **if** $L[\mathbf{y}] =$ UNLABELED **then**
25:          **if** $\exists\,\mathbf{y}' \in k\text{NN}(\mathbf{y})$ such that $L[\mathbf{y}'] = L[\mathbf{p}]$ **then**
26:             $L[\mathbf{y}] \leftarrow L[\mathbf{p}]$ {Expand current cluster label}
27:          **else**
28:             $C \leftarrow C + 1$
29:             $L[\mathbf{y}] \leftarrow C$ {New cluster due to density gap}
30:          **end if**
31:          **// Discover and schedule expansion of local neighborhood**
32:          **for** each neighbor $\mathbf{y}' \in N(\mathbf{y})$ **do**
33:             **if** $L[\mathbf{y}'] =$ UNLABELED $\wedge\, d(\mathbf{y}, \mathbf{y}') < R[\mathbf{y}']$ **then**
34:                $R[\mathbf{y}'] \leftarrow d(\mathbf{y}, \mathbf{y}')$ {Update reachability distance from the closer peak $\mathbf{y}$}
35:                $priority \leftarrow R[\mathbf{y}'] + d_k(\mathbf{y}')$
36:                $Q.\text{insert}(\mathbf{y}', \text{source} = \mathbf{y}, priority)$
37:             **end if**
38:          **end for**
39:       **end if**
40:    **end while**
41: **end for**
42: **Return** $\{L[\mathbf{x}_1], \ldots, L[\mathbf{x}_n]\}$

Algorithm 3 presents a detailed pseudocode of DANE a deterministic density-aware neighborhood propagation. To improve the efficiency of DANE, we maintains a best-so-far reachability distance from labeled points for each point:

$$R(\mathbf{x}') = \min_{L[\mathbf{x}]\neq-1\ ,\ (\mathbf{x},\mathbf{x}')\in G_k} d(\mathbf{x},\mathbf{x}')\,.$$

which guarantees that the point $\mathbf{x}'$ is inserted into the priority queue if there exists a closer predecessor of higher density. This trick significantly reduces the size of the priority queue, improving the runtime of DANE.

**Illustration of DANE.** Figure 3 shows an example of DANE process where the neighborhood of each point in $G_k$ is denoted by a circular boundary. The highest density point $\mathbf{x}_0$ creates a new cluster. Its neighbors $\mathbf{x}_4,\mathbf{x}_3,\mathbf{x}_1,\mathbf{x}_2$ are added to $Q$ with the corresponding priority score together with their predecessor $\mathbf{x}_0$. When processing $\mathbf{x}_4$, its neighbor $\mathbf{x}_3$ (and $\mathbf{x}_1$) will not be added into $Q$ as $R[\mathbf{x}_3]$ has not changed given $d(\mathbf{x}_4,\mathbf{x}_3) > d(\mathbf{x}_0,\mathbf{x}_3)$. $\mathbf{x}_5$ is added to $Q$ with a higher priority than $\mathbf{x}_2$. Though we might grow the cluster towards the direction of $\mathbf{x}_5$, by keeping the predecessor $\mathbf{x}_0$, $\mathbf{x}_2$ can still be labeled correctly. Though $\mathbf{z}_0$ might be reached by some $\mathbf{x}_i$ by some chance, it will not be labeled as its neighbors have not been labeled yet. When $Q$ is empty, the next highest density point, e.g., $\mathbf{z}_0$, is processed, creating a new cluster. Its neighbors will be added to $Q$. The process will continue until all points are labeled.

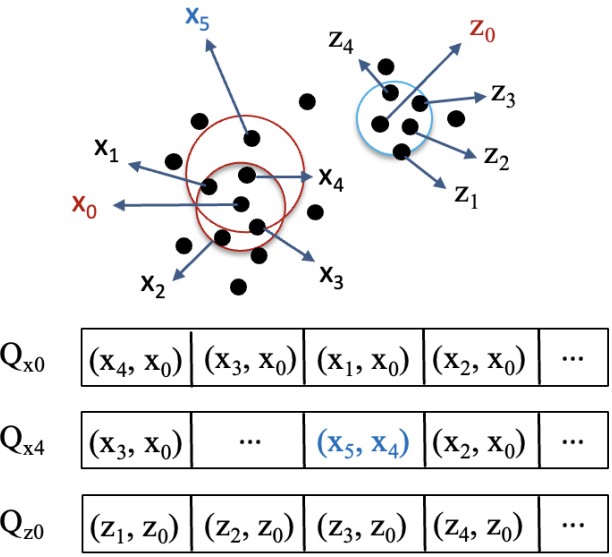

*Figure 3.* An illustration of DANE.

# B. Proof of Theorem 2

From Theorem 3.1 (Brito et al., 1997), for $k \geq c\log n$, the mutual $k$NN graph is connected within each compact, connected component of a level set $\mathcal{L}_{f_i}$ with high probability.

Given a core point $\mathbf{x} \in L_i$, we have $d_k(\mathbf{x}) \leq \varepsilon_i$. Let $V_d$ be the volume of the unit ball in $\mathbb{R}^d$, by using the $k$NN density estimator (Devroye et al., 2013):

$$\widehat{f}_k(\mathbf{x}) = \frac{k}{nV_d\left(d_k(\mathbf{x})\right)^d}\,,$$

we have $\widehat{f}_k(\mathbf{x}) \to f(\mathbf{x})$ uniformly under Lipschitz continuity. Since $\varepsilon_i < (k/nV_d f_i)^{1/d}$, any core point $\mathbf{x}$ have $\widehat{f}_k(\mathbf{x}) > f_i$, and hence $\mathbf{x} \in \mathcal{L}_{f_i}$. Alternatively, for any point $\mathbf{x}$ with density $f_0 \leq f_i < f(\mathbf{x}) < f_{i+1}$, it will be identifying as a core point in the cluster $L_i$ as $\varepsilon_{i+1} < d_k(\mathbf{x}) < \varepsilon_i$ and we run DBSCAN sequentially from $\varepsilon_\ell$ to $\varepsilon_1$.

With consistent core point sets and identical connectivity structure within each cluster (via mutual $k$NN graph and $\varepsilon_i$-neighborhood graph), the connected components formed by $DBSCAN_k^*$ and mutual $k$NN graph are equivalent with high probability as $n \to \infty$. $\qquad\square$

# C. Further Related Work

**Ordering Points for Clustering Structure (OPTICS)** (Ankerst et al., 1999) extends DBSCAN by producing an ordering of data points based on their *reachability distances*, enabling hierarchical cluster extraction across multiple density levels. Like DBSCAN, it relies on the notion of core points, but instead of forming flat clusters, OPTICS incrementally explores the data by prioritizing points that are density-reachable from known core points. Each point is assigned a reachability distance, i.e. the smallest distance needed to reach it from a core point, capturing how clusters unfold at different density scales. The resulting reachability plot reveals valleys corresponding to dense clusters, allowing users to extract meaningful clusterings without specifying a fixed global $\varepsilon$. However, OPTICS requires $O(n)$ $\varepsilon$-neighborhood search that are infeasible in large-scale high-dimensional datasets.

In scikit-learn, clusters are extracted from OPTICS using the `xi` method (Schubert & Gertz, 2018), which identifies significant changes in reachability distances to delineate dense regions. The parameter `xi` specifies the minimal steepness required in the reachability plot to recognize cluster boundaries, facilitating automatic detection of clusters across multiple density scales. While this method effectively captures clusters of varying densities, an inappropriate choice of `xi` or varied densities within a cluster may either over-segment clusters, producing fragmented results, or under-segment them, merging distinct clusters and thus compromising clustering quality.

**Label Propagation Algorithm (LPA)** (Raghavan et al., 2007) is a widely used method for graph clustering and community detection, valued for its simplicity, efficiency, and ability to uncover densely connected structures without requiring the number of clusters in advance. Starting from a set of initial labels (often randomly assigned), the algorithm iteratively updates each node's label to the most frequent label among its neighbors. This local update rule typically converges rapidly to a stable partition in a few iterations. That makes LPA well-suited for large-scale data due to its low computational overhead (e.g. $O(kn)$ in a $k$NN graph with $n$ nodes).

**Louvain and Leiden** are widely used community detection algorithms that build upon label propagation principles to identify *modular* structures in large-scale graphs. *Louvain* (Blondel et al., 2008) greedily optimizes modularity (Newman, 2006) by moving each node to the neighboring community that yields the highest modularity gain, followed by aggregation of communities into super-nodes and repeating the process hierarchically. However, Louvain can get stuck in disconnected or poorly connected communities. *Leiden* (Traag et al., 2019) addresses this by introducing a refinement phase that guarantees intra-community connectivity. It iteratively improves the community structure by locally moving nodes based on a fast local update rule and then refining the partition using a more principled stability criterion before aggregation. Both algorithms naturally support weighted graphs by incorporating edge weights into the modularity computation, allowing them to capture fine-grained community structures where edge weights encode pairwise similarities of points, particularly beneficial in $k$NN graphs derived from real-world data.

**Recent work on propagation techniques for clustering** Recent work has explored integrating Density Peak Clustering (DPC) with label propagation techniques (Li et al., 2024; Sun et al., 2025). These hybrid approaches typically identify cluster centers based on local density estimates derived from variants of the $k$NN graph and then propagate labels from high-density nodes to their neighbors. While this formulation improves cluster assignment consistency and robustness across density gradients, it critically depends on constructing a high-quality $k$NN graph that becomes computationally prohibitive for large-scale, high-dimensional datasets.

# D. Parameter settings of clustering baselines on Mnist

We detail the parameter setting of studied clustering method.

**DPC** [1]: We determine the number of clusters based on threshold values $\rho_{min}, \delta_{min} \in [0.05, 0.5]$, where both $\rho$ and $\delta$ are normalized. The cutoff distance $\varepsilon$ is chosen such that the average of $|B_\varepsilon(\mathbf{x}_i)| \in [0.01, 0.02] \cdot n$. Note that DPC requires all pairwise distances that need 30GB memory to store to run efficiently.

**DCN** [2]: We set the latent dimensionality to 10, both the number of epochs and pre-training epochs are set to 50. The regularization coefficient $\lambda$ is 0.005, and the learning rate is fixed at 0.002.

---

[1] https://github.com/pgoltstein/densitypeakclustering
[2] https://github.com/guenthereder/Deep-Clustering-Network

**DBHD** [3]: We set $\rho, \beta \in [0.3, 0.7]$ and $minClusterSize \in [4, 32]$. We replace the original kd-tree based exact $k$NN implementation with exact Faiss and observe that it speeds up the running time by $1.3\times$.

**SpectACl** [4]: We choose the target cluster number ($n\_clusters$) is 10, and set $\varepsilon$ as the smallest radius such that 90% of the points have at least 10 neighbors, with additional nearby values also considered. Besides, the embedding dimensionality is set to default value 50. This version suffers small AMI score of 45%.

**SpectACl (Normalized)** [5]: This variant replaces $\varepsilon - neighborhood$ graph with $k$NN graph to construct the normalized adjacency matrix. We vary the value of $K$ in [5, 10, 15, 20]. The target cluster number ($n\_clusters$) is also 10, and the embedding dimensionality remains at the default value of 50. This version gives 80% AMI score is used in comparison with CluProp.

**DBSCAN** [5]: We choose $\varepsilon \in [0.08, 0.12]$ (for cosine distance), and $\varepsilon \in [1000, 1300]$ (for L2 distance), $minPts \in [4, 32]$.

**OPTICS** [6]: We choose $\varepsilon$ is 0.8 for cosine distance, 2400 for L2 distance , $minPts \in [12, 32]$, $xi \in [0.001, 0.005]$.

**Kernel k-means (KKM) and spectral clustering (SC)**: Due to the limit of memory, we use the Nyström approach with 1000 samples for acceleration and for estimating the scale $\sigma$ used in the kernel.

**HDBSCAN** [7]: We use the default value $minClusterSize = 30$. HDBSCAN does not support multi-threading.

PyNNDescent: Mnist: 5 iters, leafSize = 50, 8 trees, k = 20. Pamaps: 5 iters, leafSize = 50, 16 trees, k = 20

## E. Additional experimental results on Mnist

**Other clustering metrics.** Table 5 shows comparison details between CluProp (NNDescent with Leiden) and other existing clustering on cosine distance in terms of clustering accuracy scores AMI, NMI, ARI, and CC.

*Table 5.* The clustering accuracy comparison between CluProp (NNDescent and Leiden) with $k = 8$ and other clustering baselines with cosine on Mnist. We report the maximum scores of compared baselines across various parameter settings.

| Alg. | CluProp | sDbscan | Dbscan | hDbscan | DPC | DBHD | SpectACl | SC | KKM | k-means | DCN |
|------|---------|---------|--------|---------|-----|------|----------|-----|-----|---------|-----|
| AMI (%) | **89 ± 1** | 42 ± 4 | 43 | 31 | 69 | 56 | 80 | 50 | 54 | 51 | 75 |
| NMI (%) | **89 ± 1** | 42 ± 4 | 43 | 31 | 69 | 54 | 80 | 50 | 54 | 51 | 75 |
| ARI (%) | **87 ± 1** | 9 | 9 | 5 | 54 | 18 | 70 | 37 | 41 | 38 | 62 |
| CC (%) | **88 ± 1** | 15 | 18 | 12 | 57 | 29 | 71 | 37 | 41 | 38 | 64 |

**Louvain.** Figure 4 shows the comparison in clustering accuracy between CluProp (NNDescent with Louvain) and other clustering methods. Again, we report the best AMI score after tuning their parameters. The results given by Louvain is very similar to that of Leiden over 4 studies distance measures.

**Comparison between DANE and Leiden, Louvain, LPA on PyNNDescent's graphs.** Table 6 shows the breakdown cost of CluProp with Leiden, Louvain, LPA, and DANE propagation algorithms on Mnist for with the best $k$ among $\{4, 6, \ldots, 20\}$. It is clear that DANE runs orders of magnitude faster than Leiden, Louvain, and LPA though its accuracy is approximately 10% less than modularity-based approaches. The speedup of DANE over Leiden and Louvain will be highlighted on million-point datasets Pamap2 and Mnist8m.

## F. Parameter settings of clustering baselines on Pamap2 and Mnist8m

For all density-based variants, including Dbscan, sDbscan, sngDbscan, we fix $minPts = 50$. sngDbscan uses the sampling probability $p = 0.01$.

---

[3] https://dm.cs.univie.ac.at/research/downloads
[4] https://sfb876.tu-dortmund.de/spectacl
[5] https://scikit-learn.org/stable/modules/generated/sklearn.cluster.DBSCAN.html
[6] https://scikit-learn.org/stable/modules/generated/sklearn.cluster.cluster_optics_xi.html
[7] https://hdbscan.readthedocs.io/en/latest/

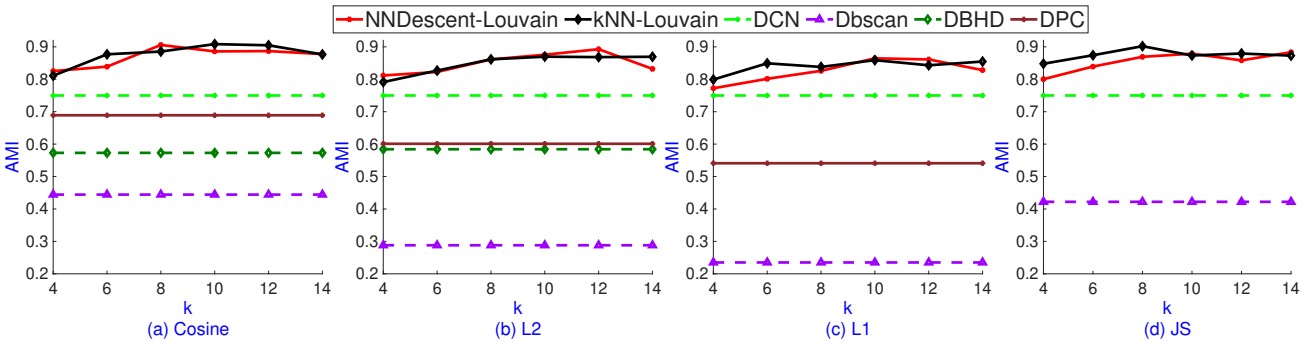

*Figure 4.* The comparison of AMI provided by CluProp (NNDescent and exact $k$NN for graph constructions with Louvain) and the maximum AMI scores provided by other clustering baselines across various parameter settings on Mnist.

*Table 6.* Comparison on label propagation algorithms: Louvain and Leiden ($k = 8$), LPA ($k = 18$) and DANE ($k = 14$) over graph constructions by PyNNDescent within 15 second.

| *Alg.* | Leiden | Louvain | LPA | DANE |
|---|---|---|---|---|
| Total time (s) | 20 | 19 | 22 | **15** |
| Clustering time (s) | 5.5 | 3.3 | 6.3 | **0.2** |
| AMI (%) | $89 \pm 1$ | $91 \pm 1$ | $77 \pm 2$ | $80 \pm 1$ |
| # clusters | $12 \pm 1$ | $12 \pm 1$ | $78 \pm 8$ | $17 \pm 2$ |

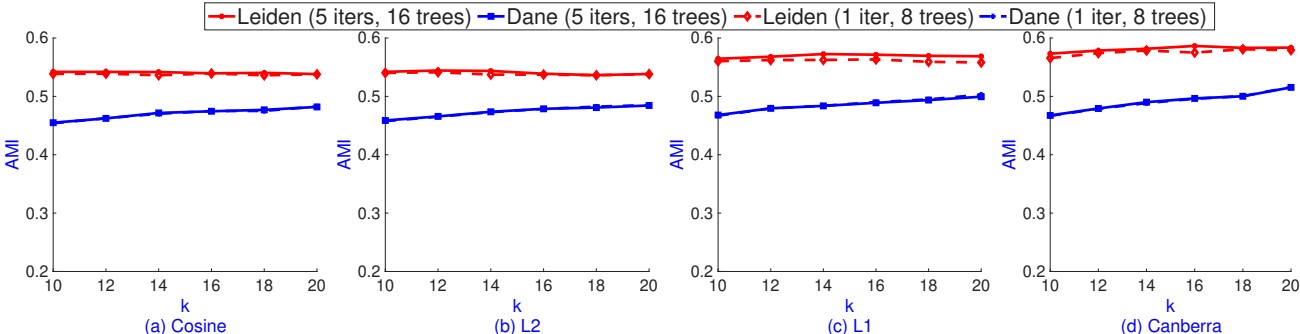

*Figure 5.* The comparison of AMI provided by CluProp with different parameters of PyNNDescent for graph constructions with Leiden and DANE across various distances on Pamap2. PyNNDescent with 5 iterations and 16 trees takes **1.4 min** while the setting of 1 iteration and 8 trees needs **0.6 min**.

On Pamaps, we select the maximum AMI scores among values of $\varepsilon$: $\varepsilon \in \{0.005, 0.01, \ldots, 0.05\}$ for cosine, $\varepsilon \in \{6, 9, \ldots, 21\}$ for L2, and $\varepsilon \in \{30, 40 \ldots, 80\}$ for L1. We also follow the suggested setting of $\sigma$ for kernel embeddings used in L1 and L2 by sDbscan (Xu & Pham, 2024).

On Mnist8m, we select the maximum AMI scores for $\varepsilon \in \{0.14, 0.15, \ldots, 0.18\}$ for cosine. Note that for $minPts = 100$, sDbscan gives similar outcomes as $minPts = 50$, as can be seen in (Xu & Pham, 2024).

## G. Additional experimental results on Pamap2

**The sensitivity of clustering accuracy on the quality of $G_k$.** Figure 5 show that the accuracy returned by Leiden and DANE is not sensitive with the quality of graph provided by PyNNDescent across many distance measures with different configurations: 5 iterations and 16 trees vs. 1 iteration and 8 trees.

## H. Additional experimental results on Mnist8m

**Leiden vs. DANE** Figure 6 show the advantage of DANE on executing on high degree graphs compared to Leiden. CluProp with DANE achieves similar AMI scores with $k = 80$ and runs 6 times faster when comparing with Leiden. Table 7 verifies the computational bottleneck of modularity-based propagation. Though PyNNDescent with $k = 50$ takes 8.2 minutes but requires 11 mintues on $k = 80$, CluProp with Leiden requires more than 1 hour but needs around 10 minutes with DANE.

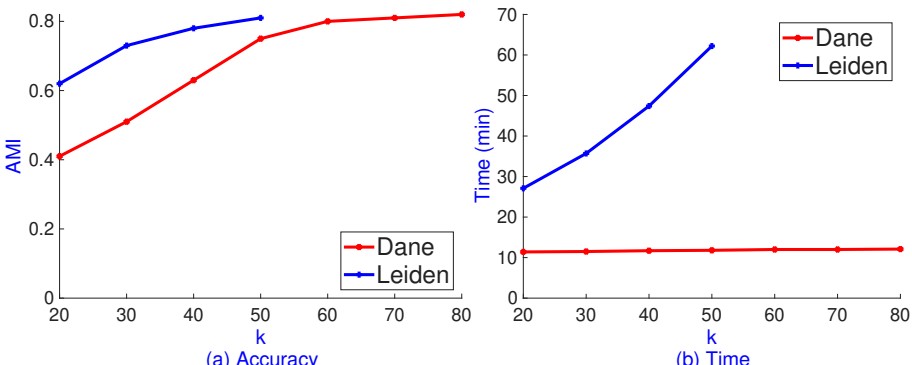

(a) Accuracy      (b) Time

*Figure 6.* The AMI and runtime of CluProp with Leiden and DANE across various values of $k$ on Mnist8m using cosine.

*Table 7.* The runtime (in minutes) comparison between Leiden and DANE across various values of $k$ on Mnist8m using cosine.

| $k$ | 20 | 30 | 40 | 50 | 60 | 70 | 80 |
|---|---|---|---|---|---|---|---|
| Leiden (min) | 19 | 27 | 39 | 54 | – | – | – |
| DANE (min) | < 1 | < 1 | < 1 | < 1 | 1 | 1 | 1 |

