# OpenReview forum: "Towards Robust and Scalable Density-based Clustering via Graph Propagation"
_ICML.cc/2026/Conference — Submitted to ICML 2026_

### Official Review · Reviewer_qwDd · 2026-03-09

**Soundness:** 3
**Presentation:** 3
**Significance:** 3
**Originality:** 3
**Overall Recommendation:** 3
**Confidence:** 5

**Summary:**

This paper proposes CluProp, which reimagines multi-density clustering in high-dimensional spaces as a label propagation process based on a neighborhood graph. It aims to mitigate the inherent parameter sensitivity of density-based methods by utilizing an efficient propagation mechanism. Experimental results, especially performance on large-scale data, demonstrate the efficiency of the proposed method.

**Compliance With Llm Reviewing Policy:**

Affirmed.

**Final Justification:**

I am very grateful to the authors for their responses to my questions in the rebuttal, some of which were well addressed. However, some concerns remain regarding this paper. For example, I feel it should be evaluated on more datasets and compared with recent algorithms, such as the SC algorithm with linear time complexity. Without these experiments, it is difficult to demonstrate whether the proposed method is effective and superior to existing algorithms.

**Key Questions For Authors:**

1. Highly varied densities (line 38, right) lack relevant paper citations.

2. Spectral clustering is a classic graph-based clustering algorithm, but no comparisons are provided. Many efficient linear algorithms exist, such as USPEC[1], which takes only 7.48 seconds on MNIST. Furthermore, newer spectral clustering methods have been developed. The method presented in this paper, however, is O(n log(n)).

[1] D. Huang, C. -D. Wang, J. -S. Wu, J. -H. Lai and C. -K. Kwoh, "Ultra-Scalable Spectral Clustering and Ensemble Clustering," in IEEE Transactions on Knowledge and Data Engineering, vol. 32, no. 6, pp. 1212-1226, 1 June 2020, doi: 10.1109/TKDE.2019.2903410.
keywords: {Clustering algorithms;Sparse matrices;Complexity theory;Robustness;Bipartite graph;Scalability;Approximation algorithms;Data clustering;large-scale clustering;spectral clustering;ensemble clustering;large-scale datasets;nonlinearly separable datasets},

3. The dataset used is too small to support the contribution and conclusion. How will it perform on datasets with millions of points?

4. The proof of Theorem 3.1 was not found. Theorem 3.2 mentions $n$ approaching infinity, but doesn't explain how it changes with $n$.

5. Table 3 shows significant differences in DANE results using different $k$ settings. How should $k$ be chosen in practical applications? and why are the results for Leiden Louvain at $k$=20 in Table 2 and at $k$=80 in Table 3 missing?

6. How are AMI and ARI calculated for algorithms involving noise (such as DBNSCAN)?

DBSCAN is sometimes uppercase and sometimes lowercase. Although the appendix discusses the parameters, a table showing the search range including the parameters of the proposed method would be clearer.

The limitations of the proposed method are not discussed.

**Limitations:**

No, due to its high time complexity, it may not be scalable for large-scale data.

**Strengths And Weaknesses:**

Strengths

The idea of ​​reimagining multi-density clustering in high-dimensional space as a label propagation process based on neighborhood graphs is very interesting. The article is well-written.  The introduction to traditional density-based methods is quite detailed.

Weaknesses

Some algorithms lacked comparisons, and the experimental setup was not very clear.

---

> ### Author Rebuttal · Authors · 2026-03-28
>
> We thanks for your reviews. We address main concerns below.
>
> __Q2. Compared to spectral clustering and USPEC__
>
> We thank for the reference. We already report two spectral clustering (SC) variants, including SpectACl (Tab 1) and Nystrom-based SC (Tab 5) with the samples p = 700. However, the comparison is not entirely like-for-like as SC requires the number of classes as an input and density-based clustering does not.
>
> Even so, USPEC paper shows NMI 67\% by USPEC and 75\% by USPEC ensemble, which are lower than NMI 80\% by SpectACl, and far below NMI 89\% by CluProp (Tab 5).
> Though USPEC and its ensemble run in 7s and 131s compared to SpectACl in 15s and CluProp in 20s, comparing the reported runtime is not adequate due to the different used hardware: Intel i5-6600 CPU 3.30 GHz with Matlab in USPEC and  AMD Ryzen Threadripper 3970X 2.2GHz with C++ & Python in CluProp.
>
> Regarding scalability, we do not think it is justified to assume that "linear" Nystrom-style SC will necessarily outperform CluProp in practice on million-point data. Their efficiency depends critically on the representative set size p: treating p as a constant is only reasonable when the low-rank approximation is sufficiently accurate, and that accuracy is itself highly parameter-sensitive. When the spectrum is not strongly skewed, larger p is needed, increasing both time and memory costs. For example, USPEC sets p = 1000 in all datasets, so the "constant" in USPEC time complexity $\sqrt{p} > \log_2{n}$ where n = 8100000 on MNIST8M.
>
> This phenomenon also occurs in Nystrom approaches for kernel k-means as we want to approximate kernel matrix K with a low-rank matrix. To predict the performance of USPEC or Nystrom-based SC on million-point MNIST8M, we should predict how large p should be used in Nystrom method to capture the eigen spectrum of the kernel matrix K.
>
> Unfortunately, real-world MNIST8M does not have such nice skewness in eigen-spectrum of the kernel matrix. The Nystrom-based kernel k-mean (Wang et al 2019) considers the sampled matrix of size n x c where c = 1600 to achieve 41\% NMI (Sec 5.2, Tab 3) within 15 min with a __supercomputer__. Wheares, CluProp and DANE use O(nd + nk) memory, achieve 80\% NMI within 12 min on a __single machine__. Given the similarity between kernel-k means and SC [1], we anticipate similar outcome when comparing to Nystrom-based SC.
>
> [1] Kernel k-means: spectral clustering and normalized cuts, KDD 04
>
> __Q3. datasets with millions of points?__
>
> We already report empirical results on two real-world million-point datasets (L310-313, left) on Sec 5.2:
>
> - Tab 2, Fig 2, Fig 5 on PAMAP2 with several distance metrics.
> - Tab 3-4, Fig 6, Tab 7 on MNIST8M on cosine metric.
>
> __Q4. Concerns of Theorem 3.1 and 3.2__
>
> Since the whole paper Brito et al. 97 is dedicated to prove Theorem 3.1, it is impossible to summarize the proof.
>
> Theorem 3.1 does not give a finite-sample threshold for n, and the results is an asymptotic scaling for k as function of n. As Theorem 3.2 uses Theorem 3.1, we cannot indicate how large n should be. We can assume n is large enough so that the size of each cluster is at least log(n), which holds in large datasets.
>
> __Q5. Table 3 shows significant differences in DANE results using different settings. How should  be chosen in practical applications? and why are the results for Leiden Louvain at =20 in Table 2 and at =80 in Table 3 missing?__
>
> The results of Leiden/Louvain at k=20 in Tab 2 is _not_ missing. You can see Leiden's accuracy in Fig 2 where we vary k in [10, 20]. Since the difference is negligible, we report Leiden/Louvain with k = 10 in Tab 2. As Leiden's runtime is proportional to the value of k, reporting Leiden at k=20 is not fair when comparing to DANE.
>
> For similar reason in Tab 3, we report k=50 as the accuracy of Leiden is stable though it runs in more than 1 hour. Fig 6 and Tab 7 show AMI scores and runtime of Leiden and DANE when increasing k. We expect to wait for nearly 2 hours if running Leiden on k=80.
>
> As DANE is 20x-60x faster than Leiden in Tab 2-3 (see Prop. Time) on the same kNN graph, we can run DANE on larger kNN graphs (community structure tends to be well-defined with larger k) to achieve comparable accuracy without sarificing the runtime.
>
> Similar to DBSCAN, we can tune k so that the number of clusters and the cluster size returned by DANE is close to our expected values. Tuning k is easier than $\epsilon$ in DBSCAN on heterogeneous data.
>
> __Q6. AMI/ARI calculated involving noise in DBSCAN?__
>
> Noise in DBSCAN is very subjective and depends on $\epsilon$ and $minPts$ values. If we treat noise as points in sparse regions like DBSCAN, DANE can detect them as small clusters at very end of the process. This is because DANE process points in priority of its density and its connectivity with dense areas. To handle noise, we can remove clusters with small size.
>
> __Reproducibility__
>
> We will specify all used parameter values of the baselines for reproducitvity.

---

> > ### Author Rebuttal · Reviewer_qwDd · 2026-04-01
> >
> > Thanks for your rebbutal.
> >
> > I still think the effectiveness of the proposed method should be demonstrated by comparing it on more datasets, and methods like USPEC should also be compared. Kernel k-means and SC are related, but not entirely equivalent, especially in experiments, where kernel k-means often performs worse than spectral clustering.

---

> > > ### Author Response · Authors · 2026-04-05
> > >
> > > We thank for your suggestion. We report new experiments to confirm the effectiveness and scalability of CluProp on additional datasets.
> > >
> > > __Scalability: New experiment with USPEC on additional datasets__
> > >
> > > We compare CluProp with USPEC and USPEC-Ens on 7 real-world datasets: Pendigits, USPS, Letters, Covertype (all from USPEC paper), MNIST, PAMAP2, MNIST8M. We use exact kNN for small datasets (Pendigit, USPS, Letters) and PyNNDescent for the rest.
> > >
> > > We test on a machine with licensed Matlab 2025b. It has AMD Ryzen 7 8845HS 3.8GHZ (8 cores, 16 threads) with 64GB RAM, a similar configuration as the one in USPEC paper to demonstrate the scalability and effectiveness of CluProp.
> > >
> > > We tune CluProp by selecting k in {4, 6, ... 20}, except million-point PAMAP2 and MNIST8M that need larger k, i.e. k = {10, 20} and {30, 80} for Leiden and DANE, respectively.
> > > While we fix #threads = 8 on CluProp, we observe that USPEC with Matlab _always_ uses > 8 threads and there is no way to control #threads in USPEC implementation. The USPEC paper did not mention the threading issue.
> > >
> > > For USPEC, we tune p = {500, 1000, ..., 3000} (p = 1000 is the default value of USPEC) and observe that increasing p does not always lead to higher accuracy, but significantly increases the runtime and memory footprint.
> > > On MNIST8M, we set p = 300 as it already took 90% of RAM at peak, and p = 400 crashes the machine.
> > > We also tune K (in KNN used in USPEC) in {4, 5, ..., 10} as suggested in the USPEC paper.
> > >
> > > For USPEC-Ens, we use ensemble size m=5 for Covertype, PAMAP2 and MNIST8M and use m=10 for the rest.
> > >
> > > We use L2 for USPEC as we observe cosine distance gives lower accuracy on some datasets (e.g. AMI in Pendigits 0.28, USPS 0.33, MNIST 0.63, MNIST8M 0.44).
> > > CluProp uses cosine on MNIST8M and L2 on the rest of the data.
> > > We do not report NMI as its score is very similar to AMI scores returned by all methods.
> > >
> > > __On small data:__ CluProp consistently returns higher accuracy than USPEC and runs faster than USPEC-Ens.
> > >
> > > |||Leiden|USPEC|USPEC-Ens|
> > > |-|-|-|-|-|
> > > |Pendigits|AMI|86 $\pm$ 1|81 $\pm$ 1|86 $\pm$ 1|
> > > ||ARI|76 $\pm$ 2|69 $\pm$ 4|79 $\pm$ 1|
> > > ||Time (s)|0.6|0.8|7.5|
> > > |USPS|AMI|76 $\pm$ 1|64 $\pm$ 1|75 $\pm$ 2|
> > > ||ARI|65 $\pm$ 3|48 $\pm$ 1|67 $\pm$ 4|
> > > ||Time (s)|1.1|1.1|11|
> > > |Letters|AMI|63 $\pm$ 1|44 $\pm$ 1|47 $\pm$ 1|
> > > ||ARI|25 $\pm$ 2|17 $\pm$ 1|20 $\pm$ 1|
> > > ||Time (s)|0.8|0.9|8.5|
> > > |MNIST|AMI|87 $\pm$ 1|70 $\pm$ 3|74 $\pm$ 2|
> > > ||ARI|86 $\pm$ 1|60 $\pm$ 6|65 $\pm$ 2|
> > > ||Time (s)|17|3.6|35|
> > >
> > > ---
> > > __On large data:__ CluProp with DANE consistently returns higher accuracy than USPEC, and running faster than USPEC-Ens and Leiden. Leiden cannot run on this machine with k $\ge$ 40 due to the memory constraint, so we report the outcome at k = 30.
> > >
> > > |||DANE|Leiden|USPEC|USPEC-Ens|
> > > |-|-|-|-|-|-|
> > > |Covertype|AMI|18 $\pm$ 0|19 $\pm$ 1|8 $\pm$ 0|10 $\pm$ 3|
> > > ||ARI|7 $\pm$ 0|7 $\pm$ 0|1 $\pm$ 0|2 $\pm$ 2|
> > > ||Time (min)|0.25|0.53|0.15|0.42|
> > > |PAMAP2|AMI|49 $\pm$ 1|54 $\pm$ 1|26 $\pm$ 4|32 $\pm$ 1|
> > > ||ARI|5 $\pm$ 1|8 $\pm$ 1|5 $\pm$ 2|17 $\pm$ 2|
> > > ||Time (min)|0.47|1.8|0.16|0.87|
> > > |MNIST8M|AMI|80 $\pm$ 1|73 $\pm$ 1|51 $\pm$ 1|55 $\pm$ 2|
> > > ||ARI|68 $\pm$ 5|47 $\pm$ 2|17 $\pm$ 1|42 $\pm$ 3|
> > > ||Time (min)|14|33|3.6|13.4|
> > >
> > > ---
> > >
> > > __Effectiveness: New experiments with other baselines on additional datasets__
> > >
> > > We compare CluProp against other density-based clustering methods (DBHD, Dbscan, Optics, hDbscan, SpecACl, DPC, DPA - suggested by reviewer ijGr) and spectral clustering (SC) on popular small UCI datasets. SC and SpecACl use #clusters as input.
> > > We use cosine distance in all data sets.
> > >
> > > We tune:
> > >
> > > - CluProp: k = {4, 6, ... 20}
> > >
> > > - Dbscan, Optics: $\epsilon$ = {0.05, 0.1, ..., 0.8}
> > >
> > > - SpectACl: k = {10, 20, ... 80}
> > >
> > > - hDbscan: minClusterSize = {5, 10, ... 50}
> > >
> > > - DPC: dc% = {0.5, 1, ..., 5}
> > >
> > > - DPA: max_k = 100.
> > >
> > > - Spectral clustering uses 'rbf' with $\gamma=1$
> > >
> > > Since these datasets are small, we use CluProp with exact kNN and Leiden. CluProp runs in under 2 seconds on all of them.
> > > The empirical results demonstrate the superiority in effectiveness of CluProp on these real-world data.
> > >
> > > |||CluProp|DBHD|Dbscan|Optics|hDbscan|SpecACl|SC|DPC|DPA|
> > > |-|-|-|-|-|-|-|-|-|-|-|
> > > |OptDigits|AMI|__93__|88|61|61|60|68|72|75|88|
> > > ||ARI|__91__|87|26|25|27|44|65|51|81|
> > > |MultiFeature|AMI|__93__|91|33|31|68|65|70|76|83|
> > > ||ARI|__93__|91|5|5|5|44|60|66|80|
> > > |Letters|AMI|__63__|61|40|39|44|37|39|31|57|
> > > ||ARI|__25__|__25__|2|2|2|5|16|10|21|
> > > |USPS|AMI|__87__|74|39|39|39|50|46|54|74|
> > > ||ARI|__80__|65|10|10|7|22|31|29|62|
> > > |PenDigits|AMI|__86__|81|68|68|71|63|66|77|85|
> > > ||ARI|76|73|48|48|50|33|50|64|__79__|
> > > |Soybean|AMI|__72__|66|44|40|57|65|69|69|54|
> > > ||ARI|38|44|18|18|30|42|__46__|__46__|24|
> > > |Semeion|AMI|__71__|66|38|38|41|47|50|48|46|
> > > ||ARI|__58__|57|1|1|1|20|35|29|26|
> > > |Dermatology|AMI|__89__|88|13|13|23|48|57|79|73|
> > > ||ARI|80|__84__|4|4|8|37|36|80|57|
> > >
> > > ---
> > >
> > > We believe these new experiments further confirm CluProp’s superior effectiveness and scalability relative to the baselines across a range of real-world datasets.

---

### Official Review · Reviewer_q9aJ · 2026-03-12

**Soundness:** 2
**Presentation:** 2
**Significance:** 2
**Originality:** 2
**Overall Recommendation:** 3
**Confidence:** 2

**Summary:**

The authors present a new clustering algorithm using a weighted KNN followed by Leiden/Louvain, and some faster variations.

**Compliance With Llm Reviewing Policy:**

Affirmed.

**Key Questions For Authors:**

- Is this algorithm truly novel? Understanding that this is always going to be hard to prove definitively, what further assurances can you provide on this point?
- What are the "true" clusters that the the algorithm is trying to recover, and what are the associated statistical guarantees?

**Limitations:**

yes

**Strengths And Weaknesses:**

To me it is a surprise that this algorithm is novel, because it is very natural:

A) Leiden/Louvain being such popular graph clustering algorithms
B) The weighted KNN graph is a very standard construction, see e.g. Tenenbaum, Joshua B., Vin de Silva, and John C. Langford. "A global geometric framework for nonlinear dimensionality reduction." Science 290.5500 (2000): 2319-2323.

As a result, I would have expected someone, at some point, to combine to the two and I'd like to hear more from the authors about how confident they could be that noone has.

There are some gaps, to me, in the literature review, e.g. anything to do with topological data analysis (including density clustering):
Wasserman, Larry. "Topological data analysis." Annual review of statistics and its application 5.2018 (2018): 501-532.
Kim, Jisu, et al. "Statistical inference for cluster trees." Advances in Neural Information Processing Systems 29 (2016).

Another issue I have - sorry if I've missed this - is a difficulty with understanding the underlying model. Suppose, for example, that the data come from a density f satisfying your conditions in Theorem 3.1. What is the interpretation of the clusters returned by CluProp in terms of f? I gather it's not as simple as the superlevel sets of f at some threshold - so what it is instead? In unsupervised problems such as clustering, I feel a clear statement of _what_ we are trying to estimate (as well as how) is important.

I get the impression the authors are also claiming that the use of _approximate_ K-NN graphs is a significant idea. Again, I'm not sure why this is not obvious.

---

> ### Author Rebuttal · Authors · 2026-03-28
>
> We thank the reviewer for raising these points and we address them below.
>
> __Q1. Novelty of the work__
>
> We agree that both weighted kNN graphs and Leiden/Louvain are individually standard tools. Our novelty claim is not that these ingredients are unimaginable in hindsight, but that this work makes a new and explicit connection between _density-based clustering_, _kNN-graph connectivity_, and _scalable graph propagation_, and then turns that connection into a practical algorithmic framework.
>
> In particular, the paper first shows that, under standard density assumptions, clusters defined as connected components of a density superlevel set can be recovered via connectivity in a mutual kNN graph, establishing an asymptotic link between varied-density $DBSCAN^*_k$ and kNN-graph connectivity.
>
> Our practical contribution is then twofold.
>
> - CluProp replaces exact mutual-kNN connectivity by propagation on an approximate weighted symmetric kNN graph, precisely because mutual kNN graphs are too fragile and too costly in the large-scale approximate regime. The paper explains that symmetric kNN preserves essential connectivity while improving robustness against fragmentation under heterogeneous densities.
>
> - DANE is not a restatement of Leiden/Louvain: it is a new density-aware propagation rule that starts from high-density points, expands labels in a density-respecting order, and uses a nearest-neighbor max-based gain rather than a modularity objective. In Sec 4.3, the paper makes this distinction explicit: modularity uses a null-model term, whereas DANE uses a max-based surrogate
> together with density-ordered processing to prevent propagation across density valleys.
>
> So the paper’s novelty is not “weighted kNN + Leiden” by itself. The contribution is: (1) a theoretical reinterpretation of varied-density DBSCAN through kNN-graph connectivity, (2) a scalable clustering framework built on approximate weighted symmetric kNN graphs, and (3) DANE, a new deterministic density-based propagation algorithm that is distinct from modularity-based community detection.
>
> __Practical impact__: When we run modularity-based graph clustering (Leiden/Louvain) on _approximate_ weighted kNN graph, we found it runs very slow on million-point data. Moreover, the approximate weighted kNN graphs often have ill-defined community structure when k is small due to fragmentation. Increasing k will increase the quality of community structure of such graph, but then Leiden will be slow down (Fig 6) and becomes the computational bottleneck of the whole clustering process (Fig 6, or Prop. time vs Total time in Tab 2-3).
>
> This is the motivation of the introduction of DANE that removes the modularity component for the speed. As DANE is 20x-60x faster than Leiden in Tab 2-3 (see Prop. Time) on the same kNN graph, we can run DANE on larger kNN graphs (community structure tends to be well-defined with larger k) to achieve comparable accuracy without sarificing the runtime (see Total time and AMI score on Tab 2-3).
>
> __Q2. Regarding the statistical target__
>
> Under the assumptions of Theorem 3.1, the population clusters are the connected components of the density superlevel set  $ { x: f(x) \geq f_0 }$. Theorem 3.2 then shows that the cluster structure induced by $DBSCAN^*_k$
>  can be recovered through mutual-kNN connectivity with high probability.
>
> Our practical methods, CluProp and DANE, should therefore be viewed as scalable algorithmic surrogates motivated by this correspondence. We agree that the present paper is stronger on this algorithmic side than on a full statistical analysis of DANE itself, and we will clarify this scope in the revision.
>
> We found that establishing a formal guarantee for DANE is nontrivial precisely because it operates on _approximate_ kNN graphs. In that setting, even well-separated clusters may be connected by erroneous approximate edges, making statistical separation difficult to control without much stronger assumptions. We will make this scope explicit in the revision.
>
> For real datasets, where the underlying density is unknown, we follow the standard ML clustering protocol and evaluate against reference labels using external metrics such as AMI, NMI, and ARI. This empirical protocol is separate from the population-level interpretation above and is intended to assess practical clustering quality on benchmark datasets.

---

> > ### Author Rebuttal · Reviewer_q9aJ · 2026-04-05
> >
> > It's a shame the authors weren't able to comment on the links to the field of topological data analysis, in which there has been considerable work on density-based clustering.
> >
> > I think the work is possibly very interesting, but it hasn't been placed very convincingly within the existing literature.
> >
> > I will keep my score as a result.

---

> > > ### Author Response · Authors · 2026-04-05
> > >
> > > We thank the reviewer for pointing out the possible connection to TDA. We agree it is an interesting background link, but we do not think it is central to the contribution of this paper, and therefore it does not justify keeping the score unchanged.
> > >
> > > Our paper is positioned against the most directly relevant literature: density-based clustering, kNN-graph connectivity, and scalable graph-based clustering. The contribution here is not topological, but algorithmic: a scalable clustering method for large, high-dimensional data, as we elaborated the novelty of our work above.
> > >
> > > More importantly, methods based on cluster-tree refinement, such as Kim et al., seem difficult to scale to our setting. Deciding whether two branches should be merged appears to require maintaining substantial geometric information between branches, which becomes a serious bottleneck on million-point datasets.
> > >
> > > So in our view, the TDA connection is complementary background rather than overlapping prior work, and it does not weaken the novelty or correctness of the paper.

---

### Official Review · Reviewer_kHVS · 2026-03-15

**Soundness:** 2
**Presentation:** 3
**Significance:** 2
**Originality:** 2
**Overall Recommendation:** 2
**Confidence:** 4

**Summary:**

This paper proposes the CluProp framework to address the scalability limitations of traditional density-based clustering methods, including DBSCAN and HDBSCAN, in high-dimensional and large-scale datasets by formulating them as graph-based community detection problems. Although this work establishes a theoretical equivalence between DBSCAN and mutual k-NN graph connectivity, their actual implementation utilizes symmetric k-NN graphs combined with modularity optimization algorithms. The framework assigns edge weights proportional to the distance between data points and leverages parallel processing through PyNNDescent for k-NN graph construction. The experimental evaluation compares the proposed method against conventional density-based baselines using AMI and ARI metrics on high-dimensional datasets, including MNIST.

**Compliance With Llm Reviewing Policy:**

Affirmed.

**Final Justification:**

The mismatch between theory and implementation remains unresolved, making the theoretical contributions questionable in practice. While prior works may justify the assumptions needed to establish the asymptotic equivalence between DBSCAN and mutual k-NN graph connectivity, they do not justify the use of symmetric k-NN in the actual implementation. Furthermore, while the additional experimental results are appreciated, they only partially address the concerns regarding the robustness and performance improvements of CluProp with DANE, as most of the evaluated datasets remain too small. Therefore, I respectfully encourage the authors to resubmit an updated version of their work to a future venue.

**Key Questions For Authors:**

1. Why is the symmetric k-NN construction used in the practical implementation theoretically justified given that the mutual k-NN assumption is needed for theoretical equivalence with DBSCAN? What would be its potential impact?

2. Given that the pipeline relies heavily on existing techniques (k-NN graphs and community detection), could you clarify the specific impact of the proposed novel modifications? Are there ablation studies to demonstrate their individual effectiveness?

3. The experimental comparison appears unfair regarding hardware utilization. What are the experimental results when all methods are evaluated using a strictly consistent thread count (e.g., all strictly single-threaded or all 32-thread)?

4. Since the framework heavily utilizes graph community detection, how does CluProp perform against other standard community detection baselines?

5. The compilation of the DANE implementation includes the -fopenmp flag. Is this implementation genuinely single-threaded in the experiments, or does it implicitly benefit from OpenMP parallelization?

6. Could you provide the exact formula used to define the edge weights during the k-NN graph construction?

**Limitations:**

No, the authors are encouraged to discuss the limitations and potential negative societal impact of this work.

**Strengths And Weaknesses:**

Strengths:

* The paper identifies and addresses the scalability limitations of classical density-based methods in high-dimensional spaces.

* The experimental results demonstrate the practical effectiveness of the symmetric k-NN graph method in clustering problems.

Weaknesses:

* It is unclear whether the symmetric k-NN construction used in the implementation is theoretically consistent with the mutual k-NN assumption underlying the theoretical equivalence with DBSCAN.

* The novelty of the proposed method is somewhat limited, as it relies on a widely used pipeline consisting of k-NN graph construction followed by community detection. While the authors introduce specific modifications, the isolated effectiveness of these novel components is not sufficiently demonstrated through ablation studies.

* The experimental evaluation appears to be unfair. The proposed CluProp framework leverages a 32-thread implementation but is compared against single-threaded baselines.

* There is no comparison with relevant community detection baselines. Because the framework fundamentally relies on graph community detection to achieve clustering, comparing it solely to density-based methods is insufficient.

* There is another potential inconsistency in experiments. Specifically, the compilation of the DANE implementation uses the -fopenmp flag, suggesting OpenMP parallelization, which contradicts the premise that the baselines are single-threaded.

* The exact mathematical definition of the edge weights for the k-NN graph construction is missing. The text states that weights are proportional to the distance between data points, but a concrete formula is not provided.

---

> ### Author Rebuttal · Authors · 2026-03-28
>
> We thanks for your reviews. We address main concerns below.
>
> __Strengths And Summary: CluProp addresses the scalability limitations of classical density-based methods__
>
> We respectfully disagree.
>
> - We address the _parameter sensitivity of density-based clustering_ on hetegeonous data by proposing a new methodology: label propagation on kNN graphs due to the connectivity and multi-density adaptivity of such graphs.
>
> - We address the _scalability of community detection algorithms_ on large kNN graphs by proposing the density-based propagation DANE.
>
> Empirical results are clear: on MNIST, Leiden/Louvain achieves 87% ARI, whereas several carefully tuned density-based baselines remain much weaker (Tab 1); and on million-point MNIST8M, DANE is faster than all DBSCAN variants while __doubling__ their AMI/NMI (Tab 3). DANE runs 50x-60x faster than Leiden/Louvain on the same kNN graphs.
>
> __W1. symmetric kNN vs. mutual kNN__
>
> As symmetric kNN connects (x, y) if x in kNN(y) OR y in kNN(x), if mutual kNN is connected, so is symmetric kNN. Therefore, Theorem 3.2 continues to hold on symmetric kNN as long as no edge links two different clusters.
>
> In theory, mutual kNN requires sufficiently large k to connect all points within the same cluster. In practice, this connectivity degrades even further when the graph is built by _approximate_ kNN search. On Mnist, mutual kNN needs k > 50 to approach accuracy of symmetric kNN with k = 6, yet still produces more than 600 clusters, directly confirming its poor connectivity.
>
> This issue becomes even more severe at scale as increasing k increases graph-construction time, which takes 90-95\% total runtime of CluProp with DANE.
>
> __W2, W4. Limited novelty compared to community detection and density-based methods__
>
> We respectfully disagree.
>
> Our main contribution is DANE, a _density-based_ propagation. DANE is inspired by density-based clustering but it is not simply a community detection algorithm: Sec 4.3 explains the key difference. DANE is 20x-50x faster than Leiden on million-point datasets (Tab 2-3) and maintains comparable clustering accuracy.
>
> Sec 5.2 is dedicated precisely to this point. It systematically compares DANE against Leiden/Louvain, SOTA community-detection baselines, on million-point datasets. Tab 2–3 report both propagation time and clustering accuracy, while Fig 6 and Tab 7 further show that Leiden slows down substantially as k increases, whereas DANE remains scalable.
>
> - __Novelty compared to Leiden (Sec 4.3)__: DANE neither optimizes modularity nor uses a null model in $E[L]$. It uses a much simpler density-based local rule, which makes it substantially faster on large graphs. Its local gain reduces to the max-based surrogate i.e. Eq (4), a fundamentally different objective and update rule from Leiden i.e. Eq (2).
>
> - __Novelty compared to DBSCAN__: DANE removes the dependence of DBSCAN on the sensitive global $\epsilon$. Instead of treating all pairs with d(x, y) < $\epsilon$ identically, DANE relies on local geometric distances on the kNN graph to propagate labels. The priority queue and predecessor mechanism ensure that clusters expand gradually from dense regions toward nearby points in __all__ directions. The connectivity check in Alg. 2 prevents accidental propagation across clusters caused by approximate kNN edges, and also helps retain very small clusters.
>
> The reason DANE gives higher accuracy than DBSCAN variants (Tab 2-3) on million-point data is not incidental: (1) kNN graphs adapt much better to varying densities than $\epsilon$-graphs, and (2) DANE exploits local geometric structure to propagate labels in a _controlled_ way.
>
> __W3. Unfair hardware setting: single-threaded vs. 32-thread__
>
> We confirm that the experimental setting is __fair__. All baselines were in 32-thread, except HDBSCAN (L731).
>
> The reported runtimes in Tab 1 are not artifacts of unequal hardware usage. SpectACl requires exact kNN graphs, DBSCAN requires exact $\epsilon$-graph, and DPC requires a full pairwise distance matrix, so their runtimes cannot plausibly be attributed to single-thread execution. DBHD is especially slow as after forming each cluster it recomputes exact kNN and $\epsilon$-neighbor over remaining unlabeled points; since it returns 118 clusters, this expensive step is repeated 118 times.
>
> __W5. -fopenmp flag with DANE__
>
> -fopenmp is used to build symmetric kNN graphs from PyNNDescent outputs. This step contributes negligibly to the total runtime: the propagation phase of DANE accounts 10\% (including < 2\% of forming the symmetric kNN graph).
>
> DANE, Leiden, and Louvain are _all_ run in single-thread (L268, right). In fact, DANE is inherently difficult to parallelize, since it starts from the highest-density point and greedily expands a cluster until no further propagation is possible; running multiple threads would break this sequential dependency (see the code src/clupig.cpp, L1606).
>
> __W6. Edge weight__
>
> We use the standard setting $w_{xy} = 1/d(x, y)$.

---

> > ### Author Rebuttal · Reviewer_kHVS · 2026-04-04
> >
> > Thank you for the detailed response. Although it addresses some of my initial questions, I still have concerns regarding this work.
> >
> > 1. The main claim of the paper relies on the theoretical equivalence between DBSCAN and mutual k-NN graph connectivity. However, if the underlying assumptions for this equivalence are violated in practice (e.g., by using symmetric k-NN), the theoretical guarantee no longer holds, making direct comparisons to density-based methods questionable. If the focus of the paper is shifting toward purely empirical state-of-the-art performance, then the baselines should also be properly optimized. For example, just as the proposed method transforms high-dimensional raw data into a k-NN graph, standard density-based methods are typically paired with dimensionality reduction (e.g., UMAP or PCA) prior to clustering in high-dimensional settings. Thus, density-based methods utilizing a dimensionality reduction step also need to be evaluated to demonstrate the empirical superiority of the proposed method.
> >
> > 2. Although DANE offers a computational advantage, it is essentially a greedy strategy based on local decisions. It lacks theoretical guarantees and depends on the quality of the underlying k-NN graph. Thus, evaluating DANE on a more diverse and complex set of datasets is necessary to reliably demonstrate its robustness and performance improvements. Three datasets (two of which are similar to each other) are insufficient for a thorough performance evaluation.
> >
> > 3. The rebuttal clarifies that 32-thread execution is used for the k-NN graph construction. However, it remains unclear to what extent the competing baselines leverage parallelization. To ensure a fair comparison, an explicit description of the parallelization techniques and settings for every baseline needs to be provided.

---

> > > ### Author Response · Authors · 2026-04-05
> > >
> > > Thank you for acknowledging that our rebuttal resolved part of your initial concerns. We address your remaining questions below.
> > >
> > > __Regarding (1)__, the assumptions used to relate mutual kNN connectivity to $DBSCAN_k^*$ are standard in prior analyses of density-based clustering (Jiang 2017; Jiang et al. 2020; Xu & Pham 2024). Without such assumptions, it would be very challenging to ensure meaningful statistical guarantee for density-based clustering.
> > >
> > > We also respectfully disagree that “dimensionality reduction + density-based clustering” is a more appropriate baseline. PCA and UMAP optimize different objectives from density-based clustering: PCA preserves global variance, while UMAP preserves local neighborhoods. Neither is designed to preserve the density-based cluster structure of the original space, so clustering after reduction can lead to fundamentally different outputs. In practice, this pipeline introduces additional hyperparameters from PCA or UMAP and can further reduce scalability, whereas our goal is precisely to minimize parameter sensitivity and improve scalability.
> > >
> > > To further demonstrate the effectiveness of CluProp, we compare against other density-based clustering methods (DBHD, Dbscan, Optics, hDbscan, SpecACl, DPC, DPA - suggested by reviewer ijGr) and spectral clustering (SC) on several popular UCI datasets. We consider cosine distance in all data sets and details of baseline parameter tuning are provided in our reply to reviewer qwDd. Since these datasets are small, we run CluProp with exact kNN and Leiden and it finishes in under 2 seconds on all datasets. The results consistently show that CluProp achieves superior clustering effectiveness on these real-world datasets.
> > >
> > > |||CluProp|DBHD|Dbscan|Optics|hDbscan|SpecACl|SC|DPC|DPA|
> > > |-|-|-|-|-|-|-|-|-|-|-|
> > > |OptDigits|AMI|__93__|88|61|61|60|68|72|75|88|
> > > ||ARI|__91__|87|26|25|27|44|65|51|81|
> > > |MultiFeature|AMI|__93__|91|33|31|68|65|70|76|83|
> > > ||ARI|__93__|91|5|5|5|44|60|66|80|
> > > |Letters|AMI|__63__|61|40|39|44|37|39|31|57|
> > > ||ARI|__25__|__25__|2|2|2|5|16|10|21|
> > > |USPS|AMI|__87__|74|39|39|39|50|46|54|74|
> > > ||ARI|__80__|65|10|10|7|22|31|29|62|
> > > |PenDigits|AMI|__86__|81|68|68|71|63|66|77|85|
> > > ||ARI|76|73|48|48|50|33|50|64|__79__|
> > > |Soybean|AMI|__72__|66|44|40|57|65|69|69|54|
> > > ||ARI|38|44|18|18|30|42|__46__|__46__|24|
> > > |Semeion|AMI|__71__|66|38|38|41|47|50|48|46|
> > > ||ARI|__58__|57|1|1|1|20|35|29|26|
> > > |Dermatology|AMI|__89__|88|13|13|23|48|57|79|73|
> > > ||ARI|80|__84__|4|4|8|37|36|80|57|
> > >
> > > We also add new experiments to compare CluProp against USPEC, a scalable spectral clustering in our reply to reviewer qwDd.
> > >
> > > We believe these new experiments further confirm CluProp’s superior effectiveness and scalability relative to the baselines across a range of real-world datasets.
> > >
> > >
> > > __Regarding (2)__, we respectfully point out that both DANE and Leiden are essentially a greedy strategy based on local decisions (via Eq (2) and (4)). None of them can have theoretical guarantees that output a clustering structure that maximizes the modularity (see Newman, Modularity and community structure in networks, PNAS 06).
> > >
> > > Both DANE and Leiden depend on the quality of the underlying k-NN graph, and we show that Leiden's scalability is reduced on large kNN graphs (see Fig 6, Tab 7). We add experiment on additional Covertype dataset (581,012 x 54) below where we tune $\epsilon$ in  {100, 110, ... 150} and minPts = 50 on DBSCAN variants.
> > >
> > > ||Leiden|DANE|DBSCAN|sDBSCAN|sngDBSCAN|
> > > |-|-|-|-|-|-|
> > > |AMI|19 $\pm$ 1|18 $\pm$ 1|8|8 $\pm$ 1|0|
> > > |ARI|7 $\pm$ 0|7 $\pm$ 0|0|1|0|
> > > |Time (min)|0.53|0.25|24|0.1|1|
> > >
> > > __Regarding (3)__, we will add details in the revision as follows. The competitors: SpectACl, DBSCAN, OPTICS, spectral clustering, DPC, and DBHD are implemented via scikit-learn functions that run in multi-threads with n_job = 32. DCN runs multi-thread via pytorch. DBSCAN variants (e.g. sngDBSCAN, sDBSCAN) uses n_threads = 32 to parallel the graph constructions.

---

### Official Review · Reviewer_ijGr · 2026-03-17

**Soundness:** 1
**Presentation:** 2
**Significance:** 1
**Originality:** 2
**Overall Recommendation:** 2
**Confidence:** 4

**Summary:**

The authors first identify DBSCAN into the framework of graph-connectivity. Then, propose a variant DBSCAN*_k where sequentially run DBSCAN into a series of increasing density thresholds by modifying $\epsilon$ while maintaining MinPts=k and prove that the cluster structure defined in this way can be recovered from the mutual k-NN graph. Based on this result, the authors propose a CluProp, a clustering method that finds the clusters as the communities of the weighted symmetric kNN graph found by approximate kNN. Finally, in order to deal with big datasets, the authors proposed DANE, where the standard community detection algorithms are substituted by a heuristic tailored to density-based clustering. Experiments in MNIST, Pamap2 and Mnist8m are provided.

**Compliance With Llm Reviewing Policy:**

Affirmed.

**Final Justification:**

While some of my concerns are solved, there are many that remain unsolved. The limitations of the method are not properly addressed anywhere (and they could be understood by using toy models and analysing the results further than by simply metrics). Moreover, the baselines are not properly explored (To properly use the oracle hyperparameter selection in HDBSCAN, one needs to tune at least the min_samples hyperparameter besides the min_cluster_size). My impression is that the method is potentially interesting, but further work is needed.

**Key Questions For Authors:**

1) Would you test your methods in other high-dimensional toy and real-world data sets?
2) Would you increase the number of baselines and carefully set their parameters?
2) Would you compare your methods with all the baselines in all these data sets?
3) What happens in intrinsically hierarchic datasets (as the case of MNIST)?

**Limitations:**

The limitations are not discussed, while several are evident (see significance), as for instance, the dependence of parameter k.

**Strengths And Weaknesses:**

Soundness:

The ideas behind DBSCAN*_k are interesting, as well as the ways to exploit its connection to graph theory. However, I see several important caveats regarding the testing and the conclusions extracted. 1) Regarding the baseline methods, the parameter space is extremely limited. and several important baselines are ignored (for instance, DPA, d'Errico et al, Information Sciences 2021). 2) The comparison with these baselines is only made on MNIST, for Mnist8m and Pamap2 the baselines are different. 3) The number of datasets is too limited to extract conclusions, I would suggest to add several toy datasets. 4) Regarding the use of MNIST, the fact that there are 10 (semantic) classes does not mean that there are 10 clusters (one can argue that a clustering method should find also different styles within each class of digit), and therefore the comparison on this dataset is not really meaningful. A deeper insight on this is needed 5) The authors include systematically DPC and DBSCAN as the same type of clustering algorithms. However, there's a fundamental difference between identify the clusters as "regions of high density separated from other regions of high density by regions of low density", as DBSCAN does, or identify the clusters as peaks of the underlying PDF (DPC, but also Mean-Shift before it). Since the theoretical results are only from DBSCAN, the inclusion of DPC in the introduction is forced.

Presentation:

The presentation shows a nice flow, going from DBSCAN to DBSCAN*_k, to CluProp to DANE. However, there are several choices that are
unclear. An example of lack of clarity is found in table I: the results for CluProp are the only ones in Bold, while the convention is to underline the best results and, in this case, hDBSCAN provides a better approximation to the number of clusters and DPC has the same number of clusters.

Significance:

Due to the limitation of the testing, the significance of the manuscript is low.

Originality:

It is worth to say that the connection between DBSCAN and graph theory is well stablished (see, for instance Schubert et al. "The relationship of DBSCAN to matrix factorization and spectral clustering." LWDA 2018-Lernen, Wissen, Daten, Analysen 2018. RWTH Aachen, 2018) and this is not recognized in the manuscript. The use of community detection algorithms to explot this connection is however, new.

---

> ### Author Rebuttal · Authors · 2026-03-27
>
> We thanks for your reviews. We address main concerns below.
>
> __W1. The parameter space is extremely limited on baselines, and several important baselines are ignored (for instance, DPA)__
>
> We respectfully disagree that the baseline search space is “extremely limited.” We state in the captions of __all__ figures/tables that we report the maximum AMI obtained by each baseline over multiple parameter settings, and the corresponding tuning protocols are detailed in Appendix D.
>
> For DPC we tune $\rho_{min}, \delta_{min} \in [0.05, 0.5]$ and the cutoff distance $\epsilon$. For DBHD, we not only tune its parameters, but also improve its runtime. For SpectACl (Hess et al., 2019), we try $\epsilon$-neighbor graph and kNN graph with several value of $k$ (L717 - 724). Our reported AMI 80\% is higher than the original one in Table 2 of SpectACl paper.
>
> Regarding DPA, we do not include it due to the _scalability_. As described in DPA, it computes the full pairwise distance matrix, which needs about 30 GB of memory on MNIST (see L708) to identify density peaks and higher-density neighbors. Note that DPA reports NMI 84\% while CluProp (Tab 5) returns 89\% on MNIST.
>
> __W2. For MNIST8M and PAMAP2 the baselines are different__
>
> This is intentional and follows directly from the paper’s scalability setting. PAMAP2 and MNIST8M are __25x and 115x__ larger than MNIST, so reusing the full MNIST baseline suite is not computationally realistic. On MNIST alone, several baselines already have substantial cost: OPTICS, HDBSCAN, DBHD, DCN, whereas CluProp runs in 20 seconds. Extrapolating such methods to million-point datasets would require many hours or more, and in some cases is simply infeasible on our hardware (e.g. DPC with 30GB of RAM for Mnist, SpectACl relies on an exact graph construction and could not finish after 2 hours on PAMAP2.)
>
> On MNIST8M, we are not aware of any result from other baselines except scalable Dbscan variants. We have to use the kernel k-means NMI score run on a supercomputer reported in Wang et al. 2019.
>
> __W3. Add toy datasets__
>
> We do not believe toy datasets would strengthen this paper. Most toy benchmarks in the clustering literature are very low-dimensional and, with careful tuning, many baselines already achieve near-perfect performance on them. In that regime, toy examples are mainly useful for visualization, not for distinguishing methods on the questions this paper targets: _scalability, robustness to heterogeneous densities, and performance on large high-dimensional real data_.
> The paper is explicitly framed around scalable clustering, emphasizing real data up to __8.1M__ points with various densities and _supporting many distance metrics_.
>
> The number of real-world datasets used in our work is comparable to prior works, e.g. Wang et al. 2019, Xu and Pham 2024, Yang et al. 2017.
>
> __W4. Regarding the use of MNIST and 10 (semantic) classes... the comparison on this dataset is not really meaningful.__
>
> We respectfully disagree. We follow the standard evaluation protocol used in much of the ML clustering literature on MNIST: reporting external clustering metrics such as AMI and ARI against the reference labels.
>
> __W5. Since the theoretical results are only from DBSCAN, the inclusion of DPC in the introduction is forced.__
>
> We respectfully disagree. Our motivation highlights a common perspective of DPC and DBSCAN: label-propagation mechanisms on geometric graphs constructed from the data. This viewpoint leads to the proposed density-based label propagation DANE designed to mitigate the parameter sensitivity of both DPC- and DBSCAN-style clustering.
>
> __W6. Due to the limitation of the testing, the significance of the manuscript is low.__
>
> We respectfully disagree.
>
> - Table 1: CluProp with Leiden significantly outperforms several popular clustering on MNIST. Though the parameters of these baselines are carefully tuned, their ARI is far from 87\% of CluProp (SpectACl: 70\%, DCN 62\%, DPC 54\%)
>
> - Table 2: CluProp with DANE outperforms recent DBSCAN variants on both time and accuracy on PAMAP2.
>
> - Table 3: CluProp with DANE outperforms recent DBSCAN variants and kernel k-means on both time and accuracy on MNIST8M. DANE __doubles__ the baselines NMI/AMI.
>
> __W7. Originality: Compared to Schubert et al. 2018__
>
> Schubert et al. considers the connectivity of the graph constructed by core points, which is similar to one of our baselines sngDbscan (Jiang et al. 2020).
>
> We consider a _harder_ version $DBSCAN^*_k$ dealing with multi-density data, governed by several parameters $\epsilon_i$ (see L157-161, left). We show that the mutual kNN graph constructed on multi-density data is still connected (Theorem 3.2). This result facilitates clustering baselines that rely exclusively on kNN graph connectivity, bypassing the need for tuning several $\epsilon_i$ on multi-density data.
>
> __W8. Parameter settings of DBSCAN variants on PAMAP2 and MNIST8M__
>
> We follow the guidance by Xu and Pham 2024 (see Appendix F)

---

> > ### Author Rebuttal · Reviewer_ijGr · 2026-04-03
> >
> > I thank the authors for their response. However, in my view the weakness raised during the review persists.
> > W1: For HDBSCAN, citing line 731 of the original manuscript: "We use the default value minClusterSize=30". I think this can hardly be described as "over multiple parameter settings". Regarding DPA, it does not need to compute the entire distance matrix but just the nearest neighbors up to a maximum.
> > W2: I understand the scalability problem, but this means that you are testing your method against the state of the art in density-based clustering just in 1 dataset.
> > W3: This directly connects to the previous point. I feel it is impossible to accept a new clustering algorithm that has only been proven to be efficient in 3 datasets, two of which are of the same data types.
> > W4: Ok, then let's say that standard metrics would not account for the whole history. At least, confusion matrices between ground truth labels and clustering results should be provided.
> > W5 While DBSCAN generates an undirected graph,  DP generates a directed one. This difference is fundamental and should be taken into consideration.
> > W6:  See reply to W3
> > W7:  Regardless of the small differences, the idea of DBSCAN as a graph is not new (and it would be desiderable that this is acknowledged in the manuscript). However, as I stated in my original review, the use of community detection algorithms to explot this connection is, to the best of my knowledge, new.

---

> > > ### Author Response · Authors · 2026-04-05
> > >
> > > We thank you for acknowledging that the use of community detection algorithms in this context is new. We address the remaining concerns below.
> > >
> > > __W1: HDBSCAN and DPA__
> > >
> > > Regarding HDBSCAN, we confirm the reported value is the best over minClusterSize in {30, 40, ..., 100}.
> > > Regarding DPA, we thank for your clarification. However, DPA still relies on exact neighborhood structure, a major bottleneck on larger datasets (see W2 for memory). Importantly, both DPC and DPA are highly fragile under approximate neighbor search because their decisions depend critically on neighbor order.
> > >
> > > - DPC: Each peak is defined through $\delta_i$, the distance to the nearest point of higher density. Non-peak point is assigned to its nearest higher-density neighbor. Missing that true neighbor under approximate search can therefore corrupt both peak identification and label assignment.
> > >
> > > - DPA: Merge decisions depend on estimated border/saddle densities between candidate peaks. If the neighborhood structure is inaccurate near thin bridges or low-density boundaries, the estimated saddle density can shift enough to change the clustering outcome.
> > >
> > > __W2: Adding Covertype__
> > >
> > > We add experiment on additional Covertype (581,012 x 54) where we tune $\epsilon$ in {100, 110, ... 150} and minPts = 50 on DBSCAN variants.
> > >
> > > ||Leiden|DANE|DBSCAN|sDBSCAN|sngDBSCAN|
> > > |-|-|-|-|-|-|
> > > |AMI|19 $\pm$ 1|18 $\pm$ 1|8|8 $\pm$ 1|0|
> > > |ARI|7 $\pm$ 0|7 $\pm$ 0|0|1|0|
> > > |Time (min)|0.53|0.25|24|0.1|1|
> > >
> > > We run DPA (via dadapy package) with k_max = 100 on Covertype. It produced more than 10K putative clusters. The corresponding saddle-point matrix alone requires 21GB of memory, and DPA did not finish within 2 hours. PAMAP2 and MNIST8M are approximately 4x and 14x larger than Covertype, respectively.
> > >
> > > __W3 and W6: More datasets and broader comparisons__
> > >
> > > We add new experiments to compare CluProp against other density-based clustering methods (including DPA) on several popular UCI datasets. Details of baseline parameter tuning are provided in our reply to reviewer qwDd. As these datasets are small, we run CluProp with exact kNN and Leiden and it finishes in under 2 seconds on all datasets. The results consistently show that CluProp achieves superior clustering effectiveness on these real-world datasets.
> > >
> > > |||CluProp|DBHD|Dbscan|Optics|hDbscan|SpecACl|SC|DPC|DPA|
> > > |-|-|-|-|-|-|-|-|-|-|-|
> > > |OptDigits|AMI|__93__|88|61|61|60|68|72|75|88|
> > > ||ARI|__91__|87|26|25|27|44|65|51|81|
> > > |MultiFeature|AMI|__93__|91|33|31|68|65|70|76|83|
> > > ||ARI|__93__|91|5|5|5|44|60|66|80|
> > > |Letters|AMI|__63__|61|40|39|44|37|39|31|57|
> > > ||ARI|__25__|__25__|2|2|2|5|16|10|21|
> > > |USPS|AMI|__87__|74|39|39|39|50|46|54|74|
> > > ||ARI|__80__|65|10|10|7|22|31|29|62|
> > > |PenDigits|AMI|__86__|81|68|68|71|63|66|77|85|
> > > ||ARI|76|73|48|48|50|33|50|64|__79__|
> > > |Soybean|AMI|__72__|66|44|40|57|65|69|69|54|
> > > ||ARI|38|44|18|18|30|42|__46__|__46__|24|
> > > |Semeion|AMI|__71__|66|38|38|41|47|50|48|46|
> > > ||ARI|__58__|57|1|1|1|20|35|29|26|
> > > |Dermatology|AMI|__89__|88|13|13|23|48|57|79|73|
> > > ||ARI|80|__84__|4|4|8|37|36|80|57|
> > >
> > > __W4. On confusion matrices.__
> > >
> > > We respectfully note that AMI, ARI, and NMI are all computed from the contingency table between predicted cluster labels and ground-truth labels. We guess by “confusion matrix” you mean this contingency table.
> > >
> > > __W5. DBSCAN and DPC as propagation mechanisms__
> > >
> > > We thank you for clarifying the distinction between undirected and directed graphs. However, at the algorithmic level, both DBSCAN and DPC still form clusters through label propagation over graph-induced neighborhood relations, which is the perspective relevant to our motivation.
> > >
> > > __W7. Relation to Schubert et al. (2018)__
> > >
> > > We will cite and acknowledge Schubert et al. (2018) in the revision. However, we respectfully disagree with the characterization that the difference is only “small.”
> > >
> > > For DBSCAN, the connection between clustering and the connectivity of the graph induced by core points is natural, since DBSCAN forms clusters by linking core points under fixed global parameters $\epsilon, minPts$. In this formulation, all core points are treated uniformly despite having different local density $|B(x, \epsilon)|$, and connections between two core points $x, y$ and $d(x, y) \le \epsilon$ are determined by the same weight.
> > >
> > > Our paper addresses a fundamental different questions: (1) how to form cluster with multi density to reduce the sensitivity of $\epsilon$, and (2) how to utilize pairwise distance in forming cluster.
> > >
> > > - For (1), we establish a new connection between $DBSCAN^*_k$ with adaptive radii $\epsilon_i$ and the connectivity of a kNN graph variant. This is conceptually and technically different from Schubert et al., who study the fixed-$\epsilon$ DBSCAN setting.
> > >
> > > - For (2), we propose DANE, together with its priority-queue and predecessor mechanism, to ensure that clusters expand progressively from dense regions toward nearby points in all directions, and to prevent erroneous propagation across clusters due to approximate kNN edges.

---

### Decision · Program_Chairs · 2026-04-30

**Decision:**

Reject

**Comment:**

The reviewers pointed out a number of issues on the submission, and all the reviewers believe that the paper is below the bar of acceptance. Given this, I would recommend reject.